# From Genetic Engineering to Preclinical Safety: A Study on Recombinant Human Interferons

**DOI:** 10.3390/ijms262411982

**Published:** 2025-12-12

**Authors:** Thelvia I. Ramos, Carlos A. Villacis-Aguirre, Emilio Lamazares, Viana Manrique-Suárez, Felipe Sandoval, Cristy N. Culqui-Tapia, Sarah Martin-Solano, Rodrigo Mansilla, Ignacio Cabezas, Oliberto Sánchez, Sergio Donoso-Erch, Natalie C. Parra, María A. Contreras, Nelson Santiago-Vispo

**Affiliations:** 1Grupo de Investigación en Sanidad Animal y Humana (GISAH), Departamento de Ciencias de la Vida y la Agricultura, Universidad de las Fuerzas Armadas ESPE, Sangolquí 171103, Ecuador; ssmartin@espe.edu.ec; 2Biotechnology and Biopharmaceutical Laboratory, Departamento de Fisiopatología, Facultad de Ciencias Biológicas, Universidad de Concepción, Víctor Lamas 1290, P.O. Box 160-C, Concepción 4030000, Chile; carlovillacis@udec.cl (C.A.V.-A.); elamazares@udec.cl (E.L.); vmanrique@udec.cl (V.M.-S.); felisandoval@udec.cl (F.S.); natparra@udec.cl (N.C.P.); mcontrerasv@udec.cl (M.A.C.); 3Carrera Ingeniería en Biotecnología, Departamento de Ciencias de la Vida y la Agricultura, Universidad de las Fuerzas Armadas—ESPE, Sangolquí 171103, Ecuador; cnculqui@espe.edu.ec; 4Laboratory of Recombinant Biopharmaceuticals, Departamento de Farmacología, Facultad de Ciencias Biológicas, Universidad de Concepción, Víctor Lamas 1290, P.O. Box 160-C, Concepción 4030000, Chile; romansilla@udec.cl (R.M.); osanchez@udec.cl (O.S.); 5Clinical Sciences Department, Faculty of Veterinary Sciences, Universidad de Concepción, Vicente Méndez 595, Chillán 3780000, Chile; oscabeza@udec.cl (I.C.); sedonoso@udec.cl (S.D.-E.); 6Clinical Biotec, 28029 Madrid, Spain; santiago@clinicalbiotec.com

**Keywords:** interferon alpha, interferon gamma, recombinant, bioactivity, antiviral, antiproliferative, immunomodulatory, preclinical, toxicity, animal model

## Abstract

There is a critical gap in the preclinical research of recombinant human interferons (rhIFNα-2b and rhIFN-γ), as most studies focus on modified variants, which complicates the understanding of the native molecules’ properties. This study addresses this limitation by comprehensively evaluating the structural stability and intrinsic toxicity of purified IFNs. Our findings confirm that both interferons retain their bioactivity (antiviral, antiproliferative, and immunomodulatory) and exhibit remarkable stability under controlled conditions. Accelerated stability assays showed that neither protein lost biological potency after 18 days at various temperatures, supporting their potential as liquid formulations. Acute and sub-chronic toxicity studies in rodent, non-rodent, and higher-organism animal models showed no signs of toxicity, even at doses 100 to 300 times higher than therapeutic levels. These assays, combined with the absence of pyrogens, support a favorable safety profile for clinical use, with no evidence of systemic or structural damage. This work establishes a reproducible experimental model and lays the groundwork for future preclinical evaluations. We underscore the importance of characterizing the safety profile of active pharmaceutical ingredients from the earliest stages of biopharmaceutical development to ensure a safe and well-founded transition to human clinical trials. Furthermore, these results open the door for the development of advanced formulations and alternative routes of administration, such as the intranasal route, an area with significant potential.

## 1. Introduction

Interferons (IFNs) are the primary effector cytokines in the innate antiviral response, differing in function, structure, and expression patterns. They are natural cell-signaling glycoproteins induced in response to viral infections, the presence of tumors, and other biological inducers [1]. IFNs constitute the first line of defense for vertebrates against infectious agents [2,3]. Scientific literature indicates that IFNs have antiviral, antiproliferative, and immunoregulatory functions that not only limit viral replication and initiate the immune response but also inhibit immunopathogenic mechanisms and minimize collateral damage from infection [3,4,5].

These cytokines are recombinant biopharmaceuticals obtained through various genetic engineering processes [2]. They are highly representative as a biotechnological product and have allowed for their application in clinical research using new therapeutic approaches [6,7]. Various formulations utilizing interferons as the active ingredient are indicated for treating diverse pathologies. However, their inherent biochemical properties have limited their therapeutic application. These biochemical properties include their protein nature, instability, molecular size, hydrophilicity, low permeability, rapid clearance from circulation, and a high susceptibility to degradation at low pH, as well as to enzymatic degradation [2,8]. These formulations require validation as biopharmaceuticals, which also includes the execution of preclinical studies to elucidate their mechanisms of action, evaluate safety and efficacy in vitro and in vivo models, and optimize therapeutic potential before their eventual application in human clinical trials [9,10,11]. Preclinical research on interferons focuses on overcoming their disadvantages as therapeutic molecules. Studies aim to mitigate adverse effects and resistance while optimizing their efficacy [12]. Preclinical studies consider new signaling pathways, identifying biomarkers to predict patient response, and developing improved analogues or new delivery strategies that maximize therapeutic benefit and reduce toxicity [2,12]. Essentially, the goal is to transform these powerful molecules into safer and more effective treatments [13,14].

Recombinant IFN production currently prioritizes Type I and II formulations, due to their capacity to activate mucosal immunity [15]. This quality positions them as ideal therapeutic candidates for first-line treatments of acute viral respiratory infections, proliferative conditions, and sexually transmitted diseases [16,17]. Currently, different formulations of Type I IFNs are available on the market. IFNα-2a and IFNα-2b are used to treat chronic hepatitis B or C, as well as various neoplasms [18,19], while IFNβ-1a and IFNβ-1b are employed in the therapy of multiple sclerosis [20]. For type II IFN, IFNγ-1b is marketed to treat chronic granulomatous disease and severe malignant osteopetrosis [21,22]. Additionally, a pharmaceutical combination of IFNα-2b and IFN-γ, used in synergistic proportions, is available and effective against basal cell carcinoma and certain viral infections [23,24].

Researchers have introduced various co-translational and post-translational modifications to the expression systems used for IFN production. These modifications focus on pharmacokinetics, which is a critical element in the clinical application of IFNs molecules. Pharmacokinetics directly influences their biological activity and the economic viability of the production process [6,25,26]. Considering these variables enabled the development of controllable, reproducible, and scalable systems [6,27,28]. Expression systems such as *Escherichia coli* and *Pichia pastoris* remain the most widely used in the production of IFNs, due to their rapid growth, ease of genetic manipulation, and ability to express proteins in large quantities [25,29,30,31]. The plasma half-lives of these molecules are short, making their prolonged presence in the body difficult (2–3 h for IFN-α, 10 h for IFN-β, and approximately 4.5 h for IFN-γ) [32,33,34]. These effects on their pharmacokinetics increase toxicity due to high and frequent doses, which affect patients’ tolerability and quality of life [35,36,37,38].

Preclinical studies of IFN-based marketed formulations have focused on evaluating their toxicity, structural modifications (such as pegylated interferons) [39,40], or combinations with other therapeutic agents [41,42]. Their therapeutic application has been restricted due to the intrinsic instability and unfavorable pharmacokinetic factors of these proteins, such as rapid systemic clearance (owing to their size and polarity) and high susceptibility to degradation by proteases and extreme pH conditions [43]. Drug delivery systems that encapsulate these proteins have been developed as transport matrices that protect the active ingredient from destabilizing interactions (such as van der Waals or electrostatic forces) and enzymatic degradation. These systems have successfully achieved an increased half-life, along with prolonged release and conserved biological activity [44]. The literature presents 22 different formulations approved by regulatory agencies, with PEGylation being the pioneering strategy in the development of these therapeutic variants [2]. Evaluated strategies include PEGylation, PASylation, self-assembling nanostructures such as liposomes and micellar systems, and micro/nanoparticles (polymeric, metallic, or hybrid) [2,45]. The successful design of these platforms requires strict consideration of rigorous criteria for safety, efficacy evaluation, manufacturing feasibility, biocompatibility, and biodegradability [46]. However, the evaluation has not focused on their active pharmaceutical ingredients [9,47] because these variants have sought to optimize molecular stability, prolong bioavailability, and mitigate systemic toxicity [43,48]. Scientists have introduced structural modifications to these molecules that alter their pharmacokinetic and pharmacodynamic properties and their immunological response [8,49]. These changes make it difficult to accurately interpret the intrinsic biological effects of these cytokines in their native form [2].

Interferons mediate their biological activity through interaction with specific membrane receptors, whose expression and affinity can vary significantly between species [50]. This interspecies specificity represents a significant limitation in extrapolating preclinical data; for example, human IFN-α has a reduced affinity for receptors in murine models [51,52]. This low affinity compromises the activation of intracellular signaling pathways, preventing the induction of representative immunological and physiological responses [51]. Consequently, the interpretation of toxicological data may be biased, with a possible underestimation of adverse events or an overestimation of the therapeutic agent’s safety profile [53]. The administration of human recombinant interferons in animal models can induce immune responses due to their recognition as exogenous proteins [54]. This immunogenicity can lead to the production of neutralizing antibodies, which interfere with the biological activity of the interferon, generating effects that are not representative of the response in humans [55]. This immune response can significantly alter pharmacokinetics, decrease bioavailability, and shorten the duration of the therapeutic effect, as interpreted from preclinical results [56,57]. Nevertheless, researchers have used various animal models to evaluate the safety profile of interferons, with numerous challenges that have made it difficult to extrapolate the results to the human context [9,58,59].

A well-designed preclinical study can be an invaluable tool for anticipating risks, optimizing dosage, and predicting clinical efficacy, thereby strengthening the basis for biopharmaceutical development [60,61]. Preclinical trials play a central role as a regulatory requirement and as a platform for understanding the basal pharmacological behavior of recombinant interferons [62]. A thorough toxicological evaluation of the active pharmaceutical ingredient is essential [63]. This knowledge is key to establishing a frame of reference, facilitating the analysis of effects, combination therapies, and failures of these recombinant drugs [64,65,66,67]. The complexity of interferons demands well-designed preclinical models that address species differences and the challenges of predicting human outcomes [53,68].

To close the gap in preclinical knowledge on recombinant interferons, this study provides a comprehensive evaluation of the intrinsic properties of two cytokine variants, rhIFN-α-2 b and rhIFN-γ. Experts thoroughly explored the potential of these recombinantly generated proteins as potent inducers of the innate immune response, with particular emphasis on their antiviral and immunomodulatory actions. This work goes beyond simple therapeutic use by assessing the functional integrity of both proteins. These findings confirm that their key biological activities, including antiviral, antiproliferative, and immunomodulatory properties, remain intact. The research also addresses the molecules’ inherent stability, demonstrating they can maintain their biological potency under various temperature conditions. The most significant impact of this work lies in the meticulous preclinical characterization. Through a comprehensive study of the safety profile of these active ingredients in rodent and non-rodent animal models, the research provides a complete understanding of their toxicity and validates their potential for development. These findings not only lay the groundwork for the creation of future, more stable, and safer formulations but also provide a guide for the preclinical evaluation of the active ingredients of biological agents. By understanding the fundamental properties of interferons in their pure state, this work facilitates a safe and effective translation of these agents from the lab to the clinic in the initial stages.

## 2. Results

The experimental development began with the genetic design of the rhIFNα-2b and rhIFN-γ constructs. The authors designed the amino acid coding sequences of both molecules, referencing NCBI notation. Structural and functional considerations previously described were incorporated to optimize their expression within heterologous systems. In the case of rhIFNα-2b, the sequence was kept identical to the human one, corresponding to a protein of 188 amino acids with an estimated molecular weight of 17 kDa. For rhIFN-γ, a spacer composed of one serine and four glycines (SGGGG) was added, followed by a six-histidine tag (His-tag) at the C-terminus, resulting in a molecular weight of approximately 17.9 kDa. The study team synthesized and cloned both sequences, which showed concordance with the reference sequences and previously described molecules in the literature.

### 2.1. Purification and Determination of the Biological Potency of the Active Ingredients rhIFNα-2b and rhIFN-γ

#### 2.1.1. Expression and Purification of Recombinant Human Interferon Alpha rhIFNα-2b

To identify the rhIFNα-2b gene cloned in the pET22b plasmid (Figure 1A), a restriction analysis was performed using NdeI and XhoI endonucleases on a 1% agarose gel (Figure 1B). A 500 bp band corresponding to the theoretical molecular weight of interferon alpha 2b was obtained (Figure 1C). Digestion with the restriction endonuclease PvuII generated a pattern that matches the theoretical prediction for the synthesized vector (Figure 1B). The 100% identity of the coding gene was confirmed by automated sequencing, as requested by the company Macrogen (https://www.macrogen.com/, accessed on 9 December 2021).

We performed an analytical induction was performed to confirm the expression of rhIFNα-2b in the *E. coli* SHuffle^®^ T7 Express strain transformed with the pET22b-rhIFNα-2b vector. Total protein samples from the induced bacteria were analyzed by 15% SDS-PAGE and Western blot assay. In both techniques, 15% SDS-PAGE and Western blot (Figure 1D), a protein band of approximately 17 kDa was identified, which was absent in the negative control and was reactive on the Western blot. The researchers normalized the results by determining the percentage of rhIFNα-2b relative to the total proteins in each sample using densitometry with ImageJ v1.54 software.

The soluble expression of the rhIFNα-2b protein was achieved through a discontinuous culture in a controlled fermenter, utilizing 5 L of TB medium. The highest expression of rhIFNα-2b was achieved nine hours post-induction, as qualitatively evaluated by 15% SDS-PAGE and Western blot (Figure 1D). A larger-scale production of rhIFNα-2b was conducted using a 5 L bioreactor under established conditions. In the first protein purification step, mixed-mode anion exchange chromatography was performed using a Capto Adhere ImpRes matrix, followed by an ultrafiltration step to prepare the input sample. The washing conditions enabled the removal of multiple contaminants, and rhIFNα-2b was the primary protein detected in the elution fraction. Weperformed two additional purification steps: cation exchange chromatography using a GigaCap S-650S matrix and high-resolution size exclusion chromatography. The protein of interest eluted with a very high purity level of 96%, as determined by Coomassie blue staining and densitometric analysis.

#### 2.1.2. In Vitro Biological Activity Assays of Purified rhIFNα-2b

##### Antiviral Activity of rhIFNα-2b

The specific antiviral activity of the purified recombinant interferons was evaluated by measuring the cytopathic effect (CPE) in HEp-2 cells exposed to the Mengo virus. The data were adjusted to a sigmoidal curve, and the half-maximal effective concentration (EC_50_) value was determined, representing the concentration that provides 50% cell protection. The rhIFNα-2b standard used to obtain the sigmoidal curve had an initial concentration of 12,000 IU/mL, with a calculated interferon titer after purification of 1.032 × 10^4^ IU/mL. The titer of the rhIFNα-2b sample was multiplied by 1/3000, the dilution of the initial sample before the assay dilutions (Figure 1E). The purified rhIFNα-2b exhibited a specific activity of 3.13 × 10^8^ IU/mg, which demonstrated its potency when compared with the standard (Figure 1E).

##### Antiproliferative Activity of rhIFNα-2b

Studies have confirmed that IFN-α can inhibit cell proliferation and induce apoptosis in certain types of cancer cells, exhibiting both antiproliferative and proapoptotic effects on HeLa cells [69]. In this experiment, we demonstrated the impact of interferon alpha on cell viability in HeLa cells through a chromogenic assay, which determines the relative fluorescence unit (RFU). The antiproliferative activity of rhIFNα-2b was determined by measuring the percentage of cell death induced in treated HeLa cells. The untreated control normalized the RFUs of the plate to express them as a percentage of cell viability. The RFU data was determined to fit a normal distribution curve. Statistical analysis established multiple comparisons between the different concentrations and the cell control, and a significant difference in cell viability was verified. Four concentrations of rhIFNα-2b significantly reduced the viability of HeLa cells: 200 ng/mL, 40 ng/mL, 8 ng/mL, and 1.6 ng/mL (F = 49.39, *p* < 0.0001). Cell death was calculated to be 46.7% for a concentration of 200 ng/mL (1:5 dilution), so the antiproliferative titer of the sample was defined as the inverse of the dilution at which 50% cell death occurred, which was greater than 200 ng/mL (Figure 1F).

Evaluation of the standard rhIFNα-2b in this same assay, revealed that four concentrations significantly reduced the viability of HeLa cells (Figure 1F): 200 ng/mL, 40 ng/mL, 8 ng/mL, and 1.6 ng/mL (F = 57.37, *p* < 0.0001). Cell death was calculated to be 46.1% for a concentration of 200 ng/mL (1:5 dilution), so the antiproliferative titer of the sample was also greater than 200 ng/mL.

#### 2.1.3. Expression and Purification of Recombinant Human Interferon Gamma rhIFN-γ

We performed a restriction analysis of this genetic construct to identify the rhIFN-γ gene cloned in the pET22b plasmid (Figure 2A), using the NdeI and XhoI endonucleases, and evaluated by 1% agarose gel electrophoresis (Figure 2B). A 500 bp band was obtained, which coincides with the theoretical molecular weight of recombinant interferon gamma (Figure 2C). Additionally, digestion with the restriction endonuclease PvuII generated a pattern that matches the theoretical prediction for the synthesized vector (Figure 2B).

The soluble expression of the rhIFN-γ protein was achieved through a batch culture in a controlled fermenter, utilizing 5 L of TB medium. We observed the highest expression of rhIFN-γ at 12 h post-induction by qualitatively evaluating the presence of rhIFN-γ using SDS-PAGE and Western blot (Figure 2D). The 15% SDS-PAGE identified that three clones of the transformed strain matched the molecular weight of 17.9 kDa. The Western blot (Figure 2D) identified a band at 17.9 kDa, which was absent in the negative control. The results were normalized, and the percentage of rhIFNα-2b relative to the total protein value in each sample was determined by densitometry (ImageJ software). Upon completion of the induction, we performed cell disruption using a bead mill (DynoMill Multilab, Willy A. Bachofen AG, Muttenz, Switzerland). An electrophoretic analysis on 15% SDS-PAGE, stained with Coomassie blue, and a Western blot indicated that most of the protein was in the soluble fraction. The protein was purified from the soluble fraction by immobilized metal-ion affinity chromatography (IMAC) (nickel) using the ÄKTA Start chromatography system (Cytiva, Marlborough, MA, USA). The process yielded a protein with 94.4% purity.

#### 2.1.4. In Vitro Biological Activity Assays of Purified rhIFN-γ

##### Antiviral Activity of rhIFN-γ

The specific activity of purified rhIFN-γ was analyzed following the same experimental design as rhIFNα-2b. The data were adjusted to a sigmoidal curve, and the EC_50_ value for purified rhIFN-γ was determined, corresponding to the dilution that generates 50% cell protection (Figure 2E). Taking the rhIFN-γ standard as a reference, with an initial concentration of 0.166 µg/mL of rhIFN-γ, a value that allowed a sigmoidal curve to be obtained and the EC_50_ to be calculated, the interferon titer was calculated with a value of 1.1 × 10^7^ IU/mL. After obtaining the titer of the rhIFN-γ sample, we multiplied the value by 1/30,000, the dilution at which the initial sample was evaluated, before performing the assay dilutions. The specific activity of the purified rhIFN-γ was 6.65 × 10^10^ IU/mg, which demonstrated the quality of the obtained interferon when compared to the standard interferon gamma, which has a specific activity of 6.64 × 10^10^ IU/mg (Figure 2E).

##### Antiproliferative Activity of rhIFN-γ

Interferon-γ plays a crucial role in activating cellular immunity and stimulating the anti-tumor immune response, which is beneficial in adjuvant immunotherapy for various types of cancer [70]. In this experiment, the effect of purified rhIFN-γ on cell viability in HeLa cells was demonstrated using a chromogenic assay that measures relative fluorescence units. We determined the antiproliferative activity of rhIFN-γ by measuring the percentage of cell death in HeLa cells treated with the recombinant human interferon gamma (rhIFN-γ). The untreated control normalized the RFUs of the plate development to express them as a percentage of cell viability. The Shapiro–Wilk test confirmed that the normalized RFU data fit a normal distribution. Using a one-way ANOVA test and a multiple comparisons test, we determined which concentrations significantly altered cell viability compared to the control. These statistical analyses revealed that four concentrations of rhIFN-γ reduced the viability of HeLa cells substantially: 200 ng/mL, 40 ng/mL, 8 ng/mL, and 1.6 ng/mL (F = 19.46, *p* ≤ 0.0001) (Figure 2F). Cell death was calculated to be 55.3% for a concentration of 200 ng/mL (1:5 dilution), and the antiproliferative titer of the sample was defined as the inverse of the dilution at which 50% cell death occurred, which was less than 200 ng/mL. For the standard rhIFN-γ, the same four concentrations significantly reduced the viability of HeLa cells (Figure 2F): 200 ng/mL, 40 ng/mL, 8 ng/mL, and 1.6 ng/mL (F = 18.24, *p* = 0.0001). Cell death was calculated to be 50.4% for a concentration of 200 ng/mL (1:5 dilution), so the antiproliferative titer of the sample was also less than 200 ng/mL.

##### Immunomodulatory Activity of rhIFN-γ

We evaluated the immunoregulatory activity of interferon in COLO-320 cells by quantifying the expression of the HLA-DR II antigen. The cells were treated for 48 h at different rhIFN-γ concentrations (0, 12.5, 25, 50, 100 ng/mL). The expression of the HLA-DR II antigen was determined at various concentrations by Western blot using the primary anti-HLA-DR II antibody and the secondary anti-mouse 680 antibody (Figure 2G). The expression of the HLA-DR II antigen in COLO-320 cells treated for 48 h showed a very similar behavior between purified rhIFN-γ and standard rhIFN-γ. The expression of the MHC class II isotype DR in COLO-320 cells was graphed in the presence of different rhIFN-γ dilutions (immunomodulatory assay) (Figure 2G). The graph confirmed the direct dependence of HLA-DR II antigen overexpression on the dose of rhIFN-γ administered to COLO-320 cells. Comparing the different rhIFN-γ concentrations of the standard protein versus the purified one reveals a significant difference at the highest concentration of 100 ng/mL, which is directly proportional to the number of antigens expressed. This demonstrates the immunoregulatory effect of the purified recombinant human interferon-γ (rhIFN-γ).

### 2.2. Stability Under Accelerated Conditions of rhIFNα-2b and rhIFN-γ

Active pharmaceutical ingredients (APIs) and drugs must be evaluated through stability studies to determine the rate of chemical and/or physical degradation that increases with temperature based on data generated under accelerated or stressed conditions, establishing a forecast value [71]. In this assay, we determined the antiviral activity of the purified rhIFNα-2b and rhIFN-γ. The protein samples were incubated at different temperatures for 18 days. The data were adjusted to a sigmoidal curve, and the EC_50_ value was calculated, corresponding to the dilution value that generates 50% cell protection (Figure 3). The results showed that the rhIFNα-2b and rhIFN-γ molecules were stable under accelerated conditions for 18 days at different temperatures. We determined the interferon titer using the IFNα-2b and IFN-γ standards as a reference, as well as the initial concentration of 12,000 IU/mL for rhIFNα-2b and 0.166 µg/mL for rhIFN-γ. We calculated the interferon titer independently and multiplied it by the dilution that was considered for the initial sample of each interferon, as previously described in the antiviral activity section (Appendix A).

The accelerated stability assay demonstrated that the specific activity of the obtained proteins remained unchanged. Performing real-time stability studies is recommended to determine how long the API remains stable at these temperatures.

### 2.3. Endotoxin Analysis (Limulus Amebocyte Lysate Test)

A linear regression model was established using known endotoxin standards ranging from 0 to 1.0 EU/mL, yielding the equation y = 0.9397x + 0.0505 with a coefficient of determination (R^2^) of 0.9836, indicating good linearity for quantitative analysis. We used this calibration curve to determine the endotoxin content of samples stored at different temperatures (4 °C, 25 °C, and 30 °C) by chromogenic LAL assay. The mean corrected absorbance value for the sample stored at 4 °C was 0.183, corresponding to an estimated endotoxin concentration of 0.14 EU/mL. In contrast, samples stored at 25 °C and 30 °C showed corrected absorbance values of −0.052, indicating no detectable endotoxin levels under these conditions. All estimated endotoxin concentrations were below the 0.2 EU/mL acceptance limit established by the United States Pharmacopeia (USP <85>) for products intended for parenteral administration, confirming compliance with regulatory standards.

### 2.4. Animal Safety Studies with the Active Ingredients rhIFNα-2b and rhIFN-γ

Characterizing the toxicity of active ingredients in vivo models is necessary to establish a safety profile, which is why alternative techniques cannot replace the use of animals. We conducted five experimental studies in animals to evaluate various parameters of the initial safety profile: acute toxicity (a median lethal dose study in mice and cardiorespiratory toxicity in rats), a pyrogen study in rabbits to evaluate possible contaminant levels of the active ingredients in an in vivo model, a subchronic toxicity assay with rhIFNα-2b administered for one month in rats, and finally, a mucosal toxicity study in sheep to determine the mucosal irritability of the active ingredients, which evaluated the treatment design and the effect on the route of administration (safety or irritability).

The human doses of IFN alpha 2b (Intron A) at 1.7 × 10^8^ IU [72,73] and for IFN gamma at 1 × 10^6^ IU [74,75] were considered.

#### 2.4.1. Median Lethal Dose Study with rhIFNα-2b and rhIFN-γ in Mice

Single-dose studies can generate useful data to describe the relationship between dose and systemic and/or local toxicity [76]. These are the initial tests that should be conducted with each substance before proceeding to other toxicity tests. Toxicological studies are most often performed with rats and mice [77]. The expression corresponds to the amount of substance administered (e.g., milligrams) per 100 g (for smaller animals) or per kilogram of the test animal’s body weight. The primary objective of acute toxicity tests is to determine the degree of toxicity of a product and its safety range, as well as the Median Lethal Dose (LD_50_), which is the dose that is lethal to 50% of the treated animals. We evaluated the mortality of the experimental group for each of the recombinant human interferons rhIFNα-2b and rhIFN-γ.

When evaluating the animals with the different doses, the existence of an LD_50_ for the interferons was not demonstrated, as no animals in the experimental groups died. This makes this study an innocuous test. The doses used were between 100 and 300 times the considered therapeutic dose (1 × 10^7^ IU, 1 × 10^6^ IU). No other signs of toxicity, such as behavioral or neurological changes, were observed. Although a product does not cause mortality at a given dose, the metabolic activity of the “target” organs may be affected, potentially inducing changes in weight and organs morphology.

The weight of the animals in each treatment group was evaluated between the beginning and the end (Table 1, Figure 4A). The paired analysis using Student’s *t*-test for dependent samples revealed significant differences in group II (t = 3.929, *p* = 0.006) and group III (t = 4.228, *p* = 0.004) corresponding to the rhIFNα-2b treatment. The administered doses of rhIFNα-2b had an influence on the weight variable. This was not the case for the control group I and the rhIFN-γ treatment group IV, which showed no significant differences in weight over time (*p* > 0.05). For both the initial and final time, no significant differences were found between the treatment groups with both interferons according to the initial weight (F = 0.952, *p* = 0.429) and also when comparing the treatment groups for the weight at the end of the study (F = 2.526, *p* = 0.078) (Table 1).

Study on organ weight post-necropsy. Macroscopic analysis:

The different organs were weighed and observed to perform the anatomo-morphological characterization of the spleen, thymus, liver, and both kidneys (Figure 4B,C). The weights of the spleen (F = 2.372, *p* = 0.92) and thymus (F = 1.400, *p* = 0.263) did not show significant differences depending on the treatment. The weights of the right kidney (F = 7.594, *p* = 0.001), the left kidney (F = 5.455, *p* = 0.004), and the liver (F = 13.766, *p* = 0.000) were significantly different between the treatment groups. This result suggests that the doses used have some influence on organs such as the liver and the kidneys. In the former, the substance undergoes first-pass metabolism and in the latter the substance is cleared.

No deaths were observed in any of the groups in the LD_50_ study due to the interferons, which makes this study an innocuity test. The animals showed no behavioral or neurological changes in any of the groups. In summary, none of the evaluated doses were toxic in the analyzed animals.

#### 2.4.2. Cardiorespiratory Toxicity with rhIFNα-2b and rhIFN-γ in Rats

The toxicity characterization of a product involves evaluating its impact on the physiology of the cardiovascular and respiratory systems. The present study aimed to determine whether high doses of rhIFNα-2b or rhIFN-γ can induce damage to both systems. We performed the evaluation of cardiorespiratory toxicity by administering single doses of 1.8 × 10^7^ IU/kg of rhIFNα-2b and rhIFN-γ, administered intraperitoneally in a volume of 150 µL, for 72 h. The basal rhythm of each animal (beats per minute, bpm, and breaths per minute, bpm) was measured, which served to compare the recordings at 1, 5, 10, 20, and 30 min after administration of rhIFNα-2b and rhIFN-γ.

During the inoculation period (single dose) and after the observation days (72 h), the animals did not die. No behavioral or neurological signs were observed. The dosage was 180 times higher than the proposed therapeutic dose for humans. Observation of heart rate and respiratory rate values revealed no differences between the groups treated with rhIFNα-2b and rhIFN-γ compared to the control (Figure 5A,B). Nor were there significant differences with respect to the control at any of the times. This essay concludes that the dose is not toxic to the cardiovascular and respiratory systems.

#### 2.4.3. Subchronic Toxicity Assays with rhIFNα-2b in Rats

Clinical therapy protocols involving interferons in humans typically require repeated administrations of high concentrations of the product. Repeated administration studies must be conducted to identify the primary toxic manifestations associated with the treatment. Although the interferon system is known to act with marked specificity for each species, these studies have documentary value regarding the already established innocuousness of interferon alpha and gamma. For this assay, subchronic toxicity, we used only one of the two IFNs, as this type of assay employs two dose levels above the selected therapeutic dose and minimizes the number of animals required. The animals in this subchronic toxicity assay were treated with three doses of rhIFNα-2b administered twice a week for 28 days. In addition, a recovery period was considered, which was evaluated in half of the animals treated with the three doses of rhIFNα-2b, and these were sacrificed 28 days post-treatment (Figure 6A). The objective of separating the animals into two periods, one month of treatment and one month post-treatment, was to determine if there was a recovery in the animals once the evaluated doses were no longer administered.

The analysis began with body and organ weights. The animals treated for 28 days exhibited inappetence, as reflected in their body weight, which was overcome in the animals 28 days post-treatment (t = −6.810, *p* = 0.000). Regarding organ weight, there was an increase in the lungs, specifically at the dose of 3 × 10^6^ IU/kg of rhIFNα-2b (Figure 6B). The livers of the animals treated with doses of 1 × 10^6^ IU/kg and 3 × 10^6^ IU/kg also showed an increase in weight. The increase in these organs was confirmed in the animals 28 days post-treatment (Figure 6C); however, there were no differences when evaluating the two time points (t = 1.354, *p* = 0.225). When comparing the two groups, no significant differences were found. The blood chemistry results (Appendix A) indicated a decrease in TGO activity in the blood, and for the two times in which the animals were divided, there were significant differences: TGO (U = 0.00, *p* = 0.029). The concentration of protein in the blood was decreased, and significant differences were found between the animals treated for 28 days and those treated for 28 days post-treatment (t = −2.879, *p* = 0.027). Another variable with substantial differences between the two times was the hematocrit (t = 3.872, *p* = 0.008). The urea concentration in the blood increased in the animals that received the dose of 3 × 10^6^ IU/kg above the value found for the control dose, but statistically, there were no significant differences (U = 2.00, *p* = 0.114).

In the case of the rest of the parameters, no significant differences were found for TGP (t = −0.376, *p* = 0.720), cholesterol (t = 2.126, *p* = 0.094), and hemoglobin (U = 11.00, *p* = 0.486). The results of the estimated variables were lower in animals sacrificed 28 days post-treatment and those that received the same scheme with three dose levels of rhIFNα-2b (Appendix A).

We conclude that, although differences were observed in weight, organ, and blood chemistry variables between the two times points, the results normalized after suspension of rhIFNα-2b administration. The effects of the biopharmaceutical on the variables had a transient effect. No serious adverse reactions or acute intolerance were observed for any of the three dose levels. The global analysis of these elements allows us to state that rhIFNα-2b was not toxic in the treated rats.

#### 2.4.4. Pyrogen Study

The pyrogen test was designed to limit an acceptable level of risk for febrile reactions related to a product or active ingredient being analyzed in animals that will be injected [78]. The assay consisted of measuring the increase in body temperature in rabbits after an intravenous injection with the sample over a determined period (Figure 7A).

When comparing temperatures across different time points between the treatment groups, no significant differences were found (ANOVA, *p* > 0.05). However, when analyzing the temperature fluctuations within each group from the start to the end of the treatment individually, we found significant differences in group V between the two time points (*p* = 0.035). For the remaining groups, no significant differences were obtained at the start (F = 2.495, *p* = 0.172) and at the end (F = 0.88, *p* = 0.536) (ANOVA, *p* > 0.05) (Appendix A).

With these results, we performed a comparison among all the animals to observe the temperature fluctuations (Figure 7B, Appendix A) and found that the two animals in the placebo group started with a higher temperature, and one of them, rabbit E, had a temperature decrease of −1.70 °C at the end of the study. Rabbits B and D treated with rhIFN-γ also initiated the first basal temperature measurements before inoculation at approximately 39 °C and exhibited a temperature decrease at the end of the study, ranging from −1.90 °C to −1.70 °C, respectively. We considered the influence of environmental temperature on those days (29 °C and 37 °C) to inform these findings. After the treatment, we observed that those animals that started the study with high temperatures between 38 °C and 39 °C managed to stabilize these temperatures, which decreased after treatment, evidence in favor of the absence of pyrogens in the active ingredients.

Another variable considered in this study was the body weight of the animal and its relationship with the treatment scheme and its doses. Three determinations were made for each of the animals. In the evaluation of body weights, there was no significant difference among the groups, as indicated by ANOVA (F = 0.017, *p* = 0.896) (Figure 7C). The animals exhibited no behavioral variation during the study.

In this study, the evaluated active ingredients met the requirements for the absence of pyrogen. No rabbit showed an individual temperature increase of 0.6 °C or more above the control temperature. No differences were observed between the groups treated with rhIFNα-2b and rhIFN-γ versus the control and placebo group for the temperature and body weight variables. The two evaluated recombinant cytokines were not toxic and remained within the normal ranges described for this test.

#### 2.4.5. Safety Study of the Active Ingredients of rhIFNα-2b-rhIFN-γ in Sheep

Several variables were considered in this trial to describe the safety effect of the active ingredients. This study focused on exploring the nasal route for future applications of these active ingredients.

Body weight was one of the first variables analyzed. Of the 16 animals studied, all 16 gained weight during the trial (Figure 8A). Upon statistical evaluation, body weight showed a normal distribution. The weight behavior during the study, as compared to the start and end of the treatment (F = 1.278, *p* = 0.396; F = 1.83, *p* = 0.195), did not show significant differences at the two time points evaluated (Appendix A).

Repeated measures ANOVA analysis demonstrated that there were no significant differences in average weights among the groups at each evaluated time point. However, when comparing the average weight at the start vs. the end in each of the treatment groups, significant differences were found in all groups: Group I (*p* = 0.013 < 0.05), Group II (*p* = 0.026 < 0.05), Group III (*p* = 0.01 < 0.05), Group IV (*p* = 0.006 < 0.05) (Appendix A). All animals experienced a weight increase at the end of the treatment, which supports the safety of the active ingredients.

Statistical analysis determined that the temperature data followed a normal distribution. When examining the average temperatures among the groups at each of the times evaluated, no significant differences were found (Figure 8B, Appendix A). When comparing the temperatures at the different weeks measured in each of the groups, there was no significant group effect or time effect, so we can conclude that there are no significant differences between the two time periods evaluated for the groups.

Another variable analyzed was related to the state of the nasal mucosa, which did not follow a normal distribution. The variable measured parameters, such as nasal secretion, obstruction, respiratory difficulty, erythema, irritation, or edema, were used to determine the physiological state of the mucosa with respect to the treatment. Some animals had nasal secretions in any of the 4 treatment groups (Group I—sheep 10; Group II—sheep 2; Group III—sheep 5, 8, and 13; Group IV—sheep 15), but each case was an isolated, single occurrence.

The histopathological study consisted of a gross evaluation of the tissue, which was performed using a nasal endoscopy (rhinoscopy). This diagnostic procedure enables direct visualization of certain internal organs and other structures within the nose, as well as the sinonasal anatomy and any associated nasal conditions. We performed weekly visualization of the nasal mucosa using rhinoscopy. No alterations were observed in any of the groups at any of the times evaluated. Several sheep exhibited melanosis, a mucosal feature characteristic of the breed type, without any other abnormalities. According to this diagnostic tool, treatment with this formulation was safe.

The histopathological study was performed specifically on the nasal mucosa, with a microscopic characterization of the animals in relation to their treatment group. Various criteria were described, such as epithelium, leukocyte infiltration, vascular congestion, and edema, which define the state of the tissue. The histopathological report revealed no differences between the treatment groups and the control group, with no lesions or damage to the nasal mucosa attributed to the treatment (Figure 8C–F). The samples revealed the presence of multiple artifacts resulting from the sampling process and animal handling, including compression necrosis, tissue crushing and deformation, and hemorrhages.

The main comparative results between both interferons are summarized in Appendix A, which shows the predominant antiviral profile of rhIFNα-2b and the characteristic immunomodulatory profile of rhIFN-γ, along with the absence of clinically relevant toxic effects in the evaluated models.

## 3. Discussion

Biopharmaceuticals, including recombinant therapeutic proteins and antibodies, have demonstrated significant potential in the treatment of various acute and chronic conditions [79]. These recombinant products offer key advantages (controlled production, high purity, and low immunogenicity) that are essential for preclinical studies evaluating efficacy, toxicity, and pharmacokinetics [80,81,82]. Recombinant bioproducts have a much more complex structure, which entails a series of special considerations in their safety and efficacy evaluation to characterize their pharmacological and toxicological profiles with an unprecedented level of precision [83]. Recombinant molecules ensure lot-to-lot purity and consistency, which is crucial for research [84,85]. This uniformity allows scientists to precisely establish dose–response relationships for both adverse and therapeutic events [86].

Within these groups of biopharmaceuticals, interferons stand out as recombinant proteins for their ability to modulate innate and adaptive immune responses, antiviral defense, immune surveillance against tumor cells, and the regulation of autoimmune responses [87]. These characteristics have positioned interferons as relevant candidates in the development of new therapeutic strategies for various pathologies [88,89]. Currently, these recombinant proteins, which can activate mucosal immunity, are presented as ideal candidates for first-line treatments for acute viral respiratory infections, proliferative conditions near the central nervous system (CNS), or sexually transmitted diseases [15]. The clinical development trajectory of recombinant interferons has demostrated only modest therapeutic efficacy, with limitations in stability, immunogenicity, and administration routes revealed in preclinical and clinical studies [82]. Despite continuous evidence demonstrating the biological potency of interferons via inhalation for treating respiratory viral infections [90,91], their clinical development has primarily focused on systemic administration (intramuscular and intravenous) to treat high-incidence and severe diseases such as hepatitis and cancer [92,93,94]. This decision diverted attention from the nasal route, ignoring its potential for a localized response with fewer side effects [95]. The history of interferons demonstrates that comprehensive preclinical research is crucial [96]. The challenges of systemic therapies have driven the search for alternative delivery systems, generating renewed interest in nasal application [2,36,97]. This demonstrates the importance of evaluating all possible routes from the earliest stages to avoid errors and maximize a biopharmaceutical’s potential [98].

Interferons are complex proteins, and any modification to their structure can affect their bioactivity [99]. The development of interferons has evolved to overcome the limitations of their protein nature [100]. Although the toxicity of modified interferons has been widely investigated [101], a comprehensive analysis of the scientific literature, encompassing over 5000 articles on recombinant interferons, revealed a significant gap in the evaluation of their intrinsic toxicity [102,103]. Most research on these cytokines has focused on lyophilized [104], conjugated [105], or modified formulations [106], such as PEGylation [107]. These formulations prolong the drug’s half-life and reduce dosing frequency, thereby optimizing pharmacokinetics and stability. However, in doing so, the introduced molecular alterations can impede the precise interpretation of the intrinsic biological effects and the safety profile of the cytokine in its native form [108]. Comprehensive API characterization is essential; preclinical findings establish the necessary safety foundation and, in turn, facilitate the future development of advanced and safe formulations, including long-acting variants, which aim to overcome the inherent pharmacokinetic limitation of the original drug [109]. Despite these innovations, the focus on formulations has made it difficult to precisely understand the biological effects attributed exclusively to the interferon molecule [110]. Challenges related to structural modifications and the use of excipients persist, which can compromise the efficacy and safety of the final product [111,112]. This situation highlights the need for future studies to evaluate the molecule in native form, enabling more accurate characterization of the molecule’s properties and therapeutic potential [2,113].

While preclinical research focuses on final formulations because they are more predictive of clinical use, this approach may have obscured relevant evidence [114]. Preclinical investigations should include comprehensive analyses of immunogenicity, tissue distribution, and systemic toxicity from both the active ingredients and their formulations to ensure a complete understanding of the drug [115]. The present study aimed to obtain, characterize the biological activity, and evaluate the stability of recombinant interferons rhIFNα-2b and rhIFN-γ as unformulated active ingredients, that is, in their native form. This characterization provides a solid foundation for the rational design of formulations, enabling the toxicity analysis of the active ingredient to determine the preclinical toxicity of the recombinant protein under study [6].

The genetic design of recombinant interferons, particularly rhIFNα-2b and rhIFN-γ, was based on a rational strategy grounded in prior knowledge of their structure and function [116,117]. For rhIFNα-2b, a sequence identical to the human version (188 amino acids) was selected to ensure structural and functional similarity to the natural molecule. This approach maximizes the probability that the protein retains the inherent biological activity of the native molecule [72]. The strategic addition of a flexible spacer (SGGGG) and a histidine tag (His-tag) at the C-terminal end modified the rhIFN-γ construct [118]. This modification not only facilitated purification steps but was also designed to avoid interfering with the protein’s bioactivity. Sequencing confirmed the integrity of both constructs, ensuring that the sequences matched published reports and reference databases (NCBI). Previously described structural and functional considerations were incorporated to optimize their expression in heterologous systems [6,119,120]. The proposed genetic design highlights the importance of rational bioengineering in the production of biopharmaceuticals [6]. By optimizing sequences for expression in heterologous systems and simplifying purification, efficient production is achieved without compromising protein functionality [121]. The methodology used was consistent with advances in recombinant interferon production and demonstrated the importance of careful design in the development of pharmaceutical products [6,27].

Success in the production of recombinant interferons depends on optimizing the expression system [122]. For this purpose, *E. coli* was selected as the host, a choice supported by its history in biopharmaceutical production and its ability to generate relatively small proteins without functional glycosylation [121,123]. In particular, the *E. coli* Shuffle^®^ T7 Express strain was engineered with mutations that favor disulfide bond formation in the cytoplasm, a critical step in the correct folding and bioactivity of interferons [122]. The gene of interest was expressed using pET series vectors, known for their high yield due to the powerful T7 promoter [124]. Previous studies have demonstrated the effectiveness of this strategy by optimizing IFN-β and IFN-γ production using similar vectors and *E. coli* strains [125,126,127]. To maximize production, key culture parameters were fine-tuned, such as medium composition, inducer concentration, and induction times. This strategy yielded biomass yields of over 1.5 g/L for both proteins, which were expressed in a soluble form in the cytoplasm [128,129]. The efficient production of recombinant interferons depends on an integrated strategy where protein solubility is a key factor [130]. The selection of an optimized host strain and the precise adjustment of culture parameters maximized yield, ensuring that the proteins were expressed in a soluble form. This simplifies purification, reduces costs, and decreases the total process time [124,131].

For the purification of recombinant interferons, strategies were implemented that reflected the initial protein design [132]. The purification of rhIFNα-2b was performed through multiple steps to achieve high purity. This sequential approach included: mixed-mode chromatography (Capto^TM^ adhere ImpRes, Cytiva, Marlborough, MA, USA), cation exchange chromatography (GigaCap S-650S, Cytiva, Marlborough, MA, USA), and high-resolution size exclusion chromatography (Sephacryl S-100 HR 26/100, Cytiva, Marlborough, MA, USA). With this combination of chromatographic steps, the protein achieved a final purity of 97%, consistent with advanced recombinant protein purification methodologies that aim to enhance process yield and efficiency by minimizing reagent consumption [133,134]. Unlike rhIFNα-2b, the genetic design of rhIFN-γ—featuring a histidine tag—significantly simplified its purification. This allowed for single-step purification via metal affinity chromatography (IMAC), a highly selective and efficient method [135]. A purity of 98.5% was achieved, a result that aligns with the findings of previous studies using the same technique [136,137]. Although this tag simplifies purification in the initial stages, its removal is recommended for clinical application by incorporating a protease cleavage site into the design to prevent potential undesirable effects [138].

The rational design, optimization of expression conditions, and purification strategy allowed for obtaining two recombinant proteins with high levels of purity and yield. The rhIFNα-2b and rhIFN-γ molecules maintained their structural integrity and biological activity after production and purification, solidifying them as stable, pharmaceutical active ingredients [139,140,141]. This enabled the characterization of the biological activity of the two active pharmaceutical ingredients (rhIFNα-2b and rhIFN-γ) according to the pharmacopeia, as antiviral, antiproliferative, and immunomodulatory [73,75].

The antiviral effect is the primary biological activity that defines rhIFNα-2b, based on gene expression control [142]. To evaluate the potency of the purified proteins, assays were performed based on their ability to inhibit the cytopathic effect of the Mengo virus in HEp-2 cells [143]. The half-maximal effective concentration of both interferons was determined, a key parameter that measures a drug’s potency [144]. The results obtained were very similar to those of the reference standards, which confirm the high quality and biological activity of the produced interferons. The demonstration of antiviral action is a crucial step in ensuring quality control in interferon production, serving as evidence of the molecule’s function [145]. Bis, et al. evaluated the antiviral activity of rhIFNα-2a against a standard (NIBSC 95/650) in a CPE assay [146]. Silega Coma, et al. determined the antiviral activity of rhIFN-γ using a standard (T-04-0511), showing that purification did not alter the biological activity [89]. The results demonstrated the similarity of the two proteins, showing that the purification process did not change the biological activity of the recombinant interferons, providing crucial evidence for the quality control and validation of the molecules’ functionality.

Scientific evidence exists regarding the anti-cancer effects of interferons by inhibiting cell proliferation, promoting cell apoptosis, and/or suppressing oncogene expression [147]. The antiproliferative activity of rhIFNα-2b and rhIFN-γ, along with their reference standards, was evaluated in the HeLa cell line using a chromogenic assay. Both interferons demonstrated similar inhibition capacities to their respective standards, confirming the antiproliferative effect of the molecules. This result was consistent with previous studies that have shown a reduction in cell proliferation with other types of interferons, such as IFN-A-2a and IFN-B-1 b [148]. Additionally, researchers have used the MTT assay to corroborate that the antiproliferative activity of these biopharmaceuticals leads to a notable decrease in cell viability [149,150,151].

The immunoregulatory activity of rhIFN-γ, characteristic of type II interferons, was evaluated by measuring its ability to induce the synthesis of the Class II HLA-DR antigen [152]. This protein is capable of causing the expression of HLA-DR genes in cells that do not normally do so, at the transcriptional level, through the class II transactivator (CIITA) [153]. The induction of the HLA-DR II antigen was confirmed by Western blot analysis in the COLO-320 and COLO-205 cell lines, demonstrating a direct relationship between the dose of rhIFN-γ and the antigen’s overexpression. These results provide evidence of the protein’s functionality, as the presence of the HLA-DR II molecule is directly linked to the IFN-γ signaling pathway [154]. The data are consistent with the literature describing the molecular regulation of CIITA and the dependence on the IFN-γ signaling pathway [153,154,155].

The evaluation of the biological activity of the interferons as active pharmaceutical ingredients demonstrated that the production and purification process did not compromise the molecules’ functionality. The cytokines remained intact throughout production, as demonstrated by the results.

The stability of biopharmaceuticals is a critical factor in determining their therapeutic efficacy, as any degradation of these highly potent molecules can compromise their biological function [156]. To ensure their quality, the stability of the active ingredients in solution was evaluated. In this study, accelerated stability tests demonstrated that both rhIFNα-2b and rhIFN-γ retained their biological potency for 18 days at various temperatures, with no significant loss of biological activity. A drug’s stability is crucial for determining its storage conditions and shelf life [157]. This stability is interrelated with toxicity, as the thermal degradation of recombinant proteins can alter their structure, increasing immunogenicity and causing undesirable inflammatory responses [131,158,159]. However, in the scientific literature, there is no direct and systematic evidence documenting the relationship between these two parameters in the context of cytokines, such as interferons [160]. While the stability of interferons does not directly correlate with toxicity, initial stability studies are necessary to determine if protein degradation under unstable environmental conditions leads to an increase in toxicity, thereby ensuring the safety of a pharmaceutical product [43].

Drug development is a progressive process in which safety evaluation serves as a decisive checkpoint before proceeding to human clinical trials [161]. To determine the safety of APIs, toxic effects, dosage, exposure, and the possible reversibility of damage are characterized [162]. The preclinical evaluation of interferons included studies of acute and subchronic toxicity, as well as mucosal irritability, in various animal models, such as mice, rats, rabbits, and sheep. These studies were conducted to predict the initial toxicity of the active ingredients. Specifically, the experiments covered five models: acute toxicity (LD_50_ in mice), cardiorespiratory toxicity in rats, subchronic repeated-dose toxicity in rats, a pyrogen test in rabbits and mucosal irritability in sheep. The preclinical safety studies conducted in this research confirmed that the recombinant interferons were safe for evaluation in subsequent studies or for use in formulations. We will first analyze the evaluation of acute and subchronic toxicity, then the pyrogen test, and finally the mucosal irritability study in sheep.

The LD_50_ did not show mortality or signs of systemic toxicity in mice, which is consistent with literature reports indicating the low acute toxicity of these biological agents [163]. Dosage in preclinical studies with mouse models is expressed in milligrams of substance per kilogram of body weight (mg/kg). This method ensures that exposure to the chemical is comparable among animals of different sizes, thereby better reflecting the actual concentration of the toxin in tissues, and toxicity is determined [164]. When evaluating the safety of a drug, regulators take the highest safe dose in animals and apply an extremely high safety factor, often 100 times or more, as was done in this study, to establish the initial Maximum Tolerated Dose (MTD) in humans [165]. This 100x margin compensates for both inherent differences between species and individual variability in human response, ensuring that any approved therapeutic dose or environmental exposure is low enough to minimize the risk of toxicity [166].

For cardiorespiratory toxicity studies in rats, regulatory principles were applied that express the dose as a quantity per unit of mass, a factor that determines actual tissue exposure [77]. A level 180 times higher than the therapeutic dose was used, which is a high dose but below that used in the LD_50_ test, which served as a maximum tolerability test. This strategy enabled the supplementation of systemic toxicity data in another murine model, with a focus on the potential effects of the recombinant on the cardiovascular and respiratory systems [167]. Despite high exposure to IFNs, none of the animals in either experiment exhibited toxicity at the organ or system levels. These results align with previous findings in the scientific literature, and confirm the favorable safety profile of both molecules even at dose levels well above the therapeutic range [9,168].

The subchronic toxicity study considered analyzing low to moderate doses ranging from 0.1 × 10^6^ to 1 × 10^6^ IU/kg/day to simulate and exceed a good margin of therapeutic exposure, as well as high doses of 3 × 10^6^ IU/kg/day to establish safety across a broad spectrum of toxicity in rats [169,170]. The subchronic toxicity study was designed to observe the toxic response, symptoms, severity, and possible reversibility of the effects caused by continuous administration of rhIFNα-2b for 28 days in rats. The results confirmed an excellent safety profile; repeated administration of the drug revealed no signs of toxicity in clinical manifestations, organ indices, or hematological and biochemical parameters in rats.

For preclinical acute toxicity studies involving recombinant IFNs in murine models, the intraperitoneal (IP) route was prioritized over intravenous (IV) or intramuscular (IM) administration, based on logistical and pharmacokinetic considerations [171]. The main advantage of the IP route lies in its ability to offer rapid systemic absorption with rich vascularization, administration of larger volumes, and effective plasma concentration, without the technical complexity of the IV route for small rodents [172]. The preclinical IP route is an effective and reproducible alternative for evaluating interferons (toxicity/efficacy), offering rapid systemic exposure with minimal animal stress [171,172].

The determination of pyrogens aimed to limit the acceptable level of risk for febrile reactions related to the products and to evaluate the most frequent adverse event described for IFNs, which is caused by traces of contaminants remaining from the production process of the molecule [173]. The results of this research confirmed that the evaluated APIs (rhIFNα-2b and rhIFN-γ) did not cause febrile reactions, meeting the standards required by regulatory agencies [174] and as established in the Convention. This result suggests that the purification process was effective in eliminating traces of contaminants. The subchronic toxicity test, which describes the animal’s response caused by continuous and repeated drug administration, is of great relevance in drug safety studies [175]. This assay included: symptoms, severity of occurrence, toxicity in primary organs, and recovery period [9]. No relevant adverse events were observed in clinical, hematological, or biochemical parameters, nor were there any morphological alterations in target organs, which is consistent with the findings reported by Rachmawati, et al. [176]. Recombinant IFNs exhibited a robust safety profile and excellent preclinical tolerability, even at doses 100–300 times the therapeutic dose, with no evidence of systemic toxicity, mortality, or irreversible organ damage [9,177]. The data obtained strengthen the safety evidence of rhIFNα-2b and rhIFN-γ, proving valuable when considering the inherent limitations in species extrapolation [178]. Although murine models are common in preclinical cytokine studies, physiological and immunological differences from humans limit the direct translation of results. This must be considered during data interpretation and the design of future clinical validation [179].

A comprehensive preclinical evaluation of biopharmaceuticals requires conducting studies in higher organisms that accurately reflect human pathophysiology. In this context, the nasal administration route was chosen for interferons based on the drug’s specific properties and high biological potency. This route is minimally invasive and allows for rapid absorption of the molecule, avoiding first-pass hepatic metabolism and increasing the biopharmaceutical’s bioavailability and efficacy [180]. The sheep was selected as an animal model because the size and surface area of its nasal epithelium make it a robust and representative model for evaluating mucosal irritability [181,182,183]. Furthermore, this is due to its anatomical and functional similarities with the human nasal system [184]. The mucosal irritability assay in sheep confirmed the safety of the recombinant interferons on the nasal mucosa. The evaluation focused on potential local damage at the administration site, with a detailed histopathological analysis that revealed no signs of inflammation, abnormal cellular infiltrate, or tissue damage, consistent with negative controls.

The evaluation in the animal models analyzed in this work demonstrated that the active ingredients of the interferons are not toxic, proposing them as viable candidates for the development of formulations. Once the safety of the active ingredients has been confirmed in preclinical models, the research should advance to a more complex phase that includes the evaluation of specific formulations [185]. However, we suggest that this research could be complemented with a transcriptomic analysis of human immune cells and the different species to quantify the differences in the toxicological response after treatment with rhIFNα-2b and rhIFN-γ. This would not only validate our hypothesis regarding attenuated signaling in rodents but also identify transcriptomic biomarkers predictive of response and toxicity for clinical trials. The next steps should involve more comprehensive preclinical assays, such as immunotoxicity, genotoxicity, carcinogenicity, and reproductive toxicity studies [186]. This methodological progression is crucial for obtaining a complete and rigorous characterization of the safety profile of the final formulations, which ensures a safe and well-founded transition to human clinical trials [187].

## 4. Materials and Methods

### 4.1. Materials

#### 4.1.1. Plasmids

pET22b: The pET-22b (+) vector (Sigma-Aldrich, St. Louis, MO, USA) features an N-terminal pelB signal sequence for potential periplasmic localization, in addition to an optional C-terminal His-Tag^®^ sequence. The sequence is numbered according to the pBR322 convention, and the T7 expression region is inverted in the circular map. The coding strand is transcribed by T7 RNA polymerase. The F1 origin is oriented to allow the generation of single-stranded DNA, corresponding to the coding strand. For single-strand sequencing, the T7 terminator primer was used.

#### 4.1.2. Bacteria and Cell Lines

*E. coli* SHuffle^®^ T7 Express (New England Biolabs, Ipswich, MA, USA). Chemically competent *E. coli* K12 cells that express the T7 promoter and are resistant to T1 phage (*fhuA2*). This is a genetically modified strain designed to promote the proper folding of proteins with multiple disulfide bonds in the cytoplasm.

HEp-2 (human laryngeal carcinoma, ATCC CCL 23): A human laryngeal carcinoma cell line that is positive for keratin (immunoperoxidase staining) and infected with HeLa cells. The ATCC confirmed that this line carries human papillomavirus DNA sequences via PCR.

HeLa (ATCC CCL-2), a human origin cell line: An immortalized epithelial cell line derived from a human cervical carcinoma.

HEp-2 and HeLa cells were grown in Dulbecco’s Modified Eagle’s Medium (DMEM). The HEp-2 culture medium was supplemented with 5% *w*/*v* fetal bovine serum (FBS), while the HeLa culture medium was supplemented with 10% *w*/*v* FBS. Neomycin (100 IU/mL) was added to both cell lines.

COLO-320 (human colorectal adenocarcinoma, ATCC CCL-220): Cell line derived from a human colon cancer biopsy.

COLO-320 cells were cultured in Roswell Park Memorial Institute medium (RPMI) supplemented with 10% FBS, glutamine (100 IU/mL), and penicillin/streptomycin (100 IU/mL). The cells were cultured at 37 °C with 5% CO_2_ and controlled humidity.

All cell lines were obtained from the American Type Culture Collection (ATCC, Manassas, VA, USA).

#### 4.1.3. Animals

In vivo studies were approved by the Bioethics and Biosecurity Committee of the Faculty of Biological Sciences at the University of Concepción (Concepción, Chile).

BALB/c mice *(Mus musculus)* from the Public Health Institute of Chile (Santiago de Chile, Chile) of both sexes, weighing between 24 and 28.5 g, for a total of 32 animals.

*Sprague Dawley* rats (*Rattus norvegicus*) from the Public Health Institute of Chile (Santiago de Chile, Chile), weighing between 300 and 400 g, for a total of 35 animals.

Adult rabbits (*Oryctolagus cuniculus)* from the Faculty of Veterinary Medicine at the University of Concepción, Chillán campus (Chillán, Chile), of the same sex, weighing more than 2.5 kg, for a total of 10 animals.

Sheep (*Ovis aries*) from the Faculty of Veterinary Medicine at the University of Concepción, Chillán campus (Chillán, Chile), female, aging 3 to 6 months and weighing 37 to 61 kg, for a total of 16 animals.

### 4.2. Methods

#### 4.2.1. Purification and Determination of the Biological Potency of the Active Ingredients rhIFNα-2b and rhIFN-γ

The base amino acid sequences for recombinant interferon alpha 2b (rhIFNα-2b) and recombinant interferon gamma (rhIFN-γ) were obtained from the NCBI (NP-000596.2, NP-000610.2). The Signal Peptide v4.1 software was used to predict the presence and location of the signal peptide, as well as to identify secretion signals. The rhIFNα-2b sequence remained identical to the human sequence, as shown in the NCBI. A serine-glycine spacer (SGGGG) along with a 6-residue histidine tag, was appended to the C-terminus of the rhIFN-γ sequence. Both sequences were optimized for expression in *E. coli* SHuffle^®^ T7 Express (a strain donated by the Laboratory of Biotechnology and Biopharmaceuticals, University of Concepción) and were synthesized by GenScript (Piscataway, NJ, USA). Cloning was performed into the pET22b vector, yielding the plasmids pET22b-rhIFNα-2b and pET22b-rhIFN-γ, which were sequenced by Macrogen Company (Seoul, Republic of Korea).

##### Expression and Purification of Recombinant Human Interferon Alpha rhIFNα-2b

Culture conditions for the expression of rhIFNα-2b in *E. coli* SHuffle^®^ T7 Express were established. Bacteria transformed with the pET22b-rhIFNα-2b plasmid were cultured in Terrific Broth (TB) medium with ampicillin (100 µg/mL) in 200 mL flasks. The Laboratory of Biotechnology and Biopharmaceuticals provided the transformed bacteria at the University of Concepción. The expression of the recombinant protein was induced under the control of the T7 promoter with isopropyl-β-D-thiogalactopyranoside (IPTG) at 1 mM by inoculating each culture into a 5 L fermenter containing TB medium supplemented with ampicillin. The bacteria were cultured at 16 °C, 25 °C, and 30 °C with agitation at 200 rpm and a pH of 7. After IPTG addition, agitation was increased to 300 rpm for 9 h. The optical density (OD) at 600 nm (OD_600_) of the biomass was measured to determine the growth kinetics. The recombinant protein, rhIFNα-2b, was identified by recognition with a primary anti-human IFN-α antibody (IFN-α Antibody (F-7) Monoclonal, Santa Cruz Biotechnology Inc., Dallas, TX, USA) with an incubation time of 2 h at room temperature and a concentration of 0.2 µg/mL. The reaction with the secondary anti-mouse IgG antibody conjugated to Alexa Fluor 680 (Jackson ImmunoResearch Inc., West Grove, PA, USA) was incubated for 1 h at room temperature at a concentration of 0.1 µg/mL. The cells were collected by centrifugation after induction and resuspended in buffer (50 mM Tris/HCl, 0.5 mM PMSF, pH 7.5) at a ratio of 40 mL per gram of wet biomass obtained from the fermentation, at a concentration of 100 mg/mL. The cells were homogenized at high pressure (EmulsiFlex C5, Avestin Inc., Ottawa, ON, Canada) in 9 passes at 1500 psi. The lysate was diluted to 25 mg/mL and then centrifuged at 17,020× *g* for 20 min at 4 °C, separating the soluble and insoluble fractions. 0.1 mM PMSF was added to both fractions. The soluble fraction was clarified by a second centrifugation (13,790× *g*, 5 min, 4 °C) and then filtered through 0.45 µm and 0.22 µm membranes. The samples were analyzed by 15% sodium dodecyl sulfate-polyacrylamide gel electrophoresis (SDS-PAGE) and Western blot, using a primary anti-human IFN-α antibody (IFN-α Antibody (F-7) Monoclonal, Santa Cruz Biotechnology Inc., Dallas, TX, USA) and a secondary anti-mouse IgG antibody conjugated to Alexa Fluor 680 (Jackson ImmunoResearch Inc., West Grove, PA, USA). A concentration and removal step for proteins larger than 100 kDa was performed using tangential ultrafiltration (Sartorius Sartocon^®^ Slice 200 Holder, Sartorius, Göttingen, Germany) equipped with a 100 kDa polyethersulfone (PESU) membrane. The membrane was equilibrated with 50 mM Tris/HCl buffer, pH 8, while processing 800 mL of the soluble fraction at a flow rate of 50 mL/min, with 30 psi in the retentate and 10 psi in the permeate. The retained material was diluted four times in 50 mM Tris/HCl buffer, 0.1 mM PMSF, pH 8, and then centrifuged at 17,020× *g* for 20 min. The sample was concentrated to approximately 100 mL, washed with the dilution buffer, and analyzed by 15% SDS-PAGE and Western blot. It was then stored at −20 °C.

The chromatographic purification of the recombinant protein was performed in three sequential stages. It began with mixed-mode interaction chromatography using an XK 50/20 column packed with 100 mL of Capto Adhere ImpRes resin (Cytiva, Marlborough, MA, USA), which had been previously equilibrated with 50 mM Tris-HCl buffer containing 0.1 mM PMSF, pH 8. The sample was loaded at a linear velocity of 76 cm/h and subsequently washed with the same buffer. Contaminating proteins were removed by washing with Tris-HCl buffer supplemented with 300 mM NaCl. The elution of rhIFNα-2b was performed using a 50 mM citrate buffer, 200 mM NaCl, 0.1 mM PMSF, and a pH of 4.3. Afterward, a cationic exchange stage was conducted using an XK 50/20 column with 100 mL of GigaCap S-650s matrix (Cytiva, Marlborough, MA, USA), equilibrated with 50 mM citrate buffer containing 0.1 mM PMSF at pH 4.3. The sample, previously diluted in the same buffer, was applied to the column, and the elution was carried out using 50 mM citrate buffer with 300 mM NaCl, pH 4.3. The eluting fraction containing the protein of interest was diafiltered in 50 mM sodium phosphate buffer, pH 7.4. Finally, size-exclusion High-Performance Liquid Chromatography (SEC-HPLC) was performed using a 26/100 column packed with Sephacryl S-100 HR resin (Cytiva, Marlborough, MA, USA), which was equilibrated with 100 mM phosphate buffer and 300 mM NaCl at pH 7.5. All fractions (soluble, insoluble, and purified) were evaluated by 15% SDS-PAGE and Western blot. The elution peaks obtained were collected and subsequently analyzed to confirm the purity of the recombinant protein. All samples were stored at −20 °C.

##### Expression and Purification of Recombinant Human Interferon Gamma rhIFN-γ

The recombinant strain *E. coli* SHuffle^®^ T7 Express transformed with the pET22b-rhIFN-γ plasmid (donated by the Laboratory of Biotechnology and Biopharmaceuticals at the University of Concepción) was cultured in Terrific Broth (TB) medium supplemented with ampicillin (100 µg/mL) in a 10 L fermenter at 30 °C until an OD_600_ of 0.9 was reached. The expression of the recombinant protein, under the control of the T7 promoter, was induced with IPTG at a final concentration of 0.5 mM, followed by a decrease in temperature to 20 °C and a 12 h induction period. The collected cells were subjected to cell disruption by mechanical homogenization (VELP Scientific Homogenizer, VELP Scientifica Srl, Usmate, Italy) in a lysis buffer composed of 50 mM sodium succinate (8.1 g/L), 40 g/L mannitol, pH 5. To improve process efficiency, cell disruption was performed using a 0.4 mm glass bead mill with a retention time of 4.8 min per pass (DynoMill Multilab, Willy A. Bachofen AG, Muttenz, Switzerland). Subsequently, the soluble and insoluble fractions were separated by centrifugation at 17,020× *g* for 1 h at 15 °C. The samples were analyzed by 15% SDS-PAGE and Western blot using a monoclonal anti-His antibody and a secondary anti-mouse IgG antibody conjugated to Alexa Fluor 680 (Jackson ImmunoResearch Inc., West Grove, PA, USA). The samples were then stored at −20 °C.

The purification of rhIFN-γ was performed from the soluble fraction by metal ion affinity chromatography (IMAC), using the ÄKTA Start system (Cytiva, Marlborough, MA, USA). The sample was diluted 1:1 with equilibration buffer (50 mM phosphate, 150 mM NaCl, 25 mM imidazole, pH 7.4) and applied to a 72 mL Chelating Sepharose Fast Flow (FF) (Cytiva, Marlborough, MA, USA) column loaded with nickel sulfate. Recirculation was maintained for 4 h at 2 mL/min. After washing with buffer (50 mM phosphate, 150 mM NaCl, 200 mM imidazole), elution was performed with elution buffer (50 mM phosphate, 150 mM NaCl, 500 mM imidazole, 10% glycerol, pH 7.4). Buffer exchange was performed to remove imidazole down to 500 µg/mL. The final buffer contained 50 mM sodium succinate and 40 g/L mannitol, at a pH of 5. Additionally, two passes were applied in PBS for diafiltration using a Pellicon XL cassette (5 kDa), as described in the procedure reported for rhIFNα-2b, utilizing the Sartocon Slice 200 Holder system (Sartorius, Göttingen, Germany) at a flow rate of 45 mL/min. The sample was diafiltered against 600 mL of buffer and concentrated to 25 mL. Glycerol was added at 8.5% as a stabilizer.

The quantification of total proteins was performed using the Bradford method, with BSA as a standard (2 mg/mL), and the absorbance was measured at 595 nm on a Synergy HTX Multimode Reader (Agilent Technologies Inc., Santa Clara, CA, USA). Confirmation of rhIFN-γ expression was performed using 15% SDS-PAGE and Western blot, as described previously. The transfer was verified with 1% Ponceau staining and protein detection was performed using a monoclonal anti-his antibody (Jackson ImmunoResearch Inc., West Grove, PA, USA) and a secondary anti-mouse IgG antibody conjugated to Alexa Fluor 680 (Jackson ImmunoResearch Inc, West Grove, PA, USA), developed on the Odyssey CLx Imaging System (LI-COR Biosciences, Lincoln, NE, USA).

#### 4.2.2. In Vitro Biological Activity Assays

##### Antiviral Activity of Purified rhIFNα-2b and rhIFN-γ

Antiviral activity of recombinant interferons alpha-2b (rhIFNα-2b) and gamma (rhIFN-γ) was analyzed based on the ability of the interferons to prevent viral infection and inhibit the cytopathogenic effect (CPE) produced by the Mengo virus [188,189] on the HEp-2 cell line. The potency of the obtained interferons was estimated by comparing their protective action against the commercial reference interferons [145].

HEp-2 cells were seeded in 96-well plates (Corning Costar, Thermo Fisher Scientific, Waltham, MA, USA) at 1.5 × 10^4^ cells/well in DMEM + 5% FBS + neomycin (100 IU/mL) and incubated at 37 °C and 5% CO_2_ for 24 h. Cells were treated with different concentrations of rhIFNα-2b and rhIFN-γ for 24 h at 37 °C and 5% CO_2_. For the commercial interferons rhIFNα-2b [190] and rhIFN-γ [191], eight serial dilutions were prepared with a factor of 1:5 in DMEM + SFB 2%, starting at 12,000 IU/mL, for the purified rhIFNα-2b and rhIFN-γ. Eight serial dilutions were made in DMEM + SFB 2% with a factor of 1:5, starting with a dilution of 1/3000 from a solution at 500 µg/mL. The medium was replaced with 100 µL of Mengo virus in DMEM supplemented with 2% FBS, and the plates were incubated at 37 °C with 5% CO_2_ for 24 h. The plates were washed, fixed, and stained for 15 min with 0.5% crystal violet solution in 20% methanol. The crystal violet was dissolved with 10% acetic acid, and the plate absorbance reading was performed at 590 nm on a Synergy HTX Multimode Reader spectrophotometer (Agilent Technologies Inc., Santa Clara, CA, USA). The data were fitted to a sigmoidal curve to determine the median effective concentration, defined as the dilution that generates 50% cell protection after a specified time of exposure and is indicative of potency [188,189].

Cell control (CC) wells with cells without Mengo virus were designated as cell control (CC), and cells without interferon treatment exposed to the infectious agent were designated as virus control (VC). The EC_50_ of a quantal dose–response curve represents the concentration of a compound at which 50% of the population shows a response. The EC_50_ value was determined using Equation (1) to fit the data, calculating the potency of the interferons in comparison to the commercial standards.(1)$$Abs\;norm=\frac{Abs-cv}{cc-cv}$$

Considering this, the interferon titer was calculated according to Equation (2):(2)$$IFN\;titer\;\left(\frac{IU}{mL}\right)=\frac{Sample\;titer}{STD\;titer}\times STD\;\left(\frac{IU}{mL}\right)$$
and the specific activity with Equation (3):(3)$$Specific\;activity\;\left(\frac{IU}{mL}\right)=\frac{IFN\;titer\;(IU/mL)}{IFN\;concentration\;(mg/mL)}$$

The data were fitted to a sigmoid curve, which determined the value of the EC_50_—the dilution that generated 50% cell death—and the titer and specific activity of the purified interferons.

##### Antiproliferative Activity of Purified rhIFNα-2b and rhIFN-γ

Antiproliferative activity of rhIFNα-2b and rhIFN-γ was evaluated in HeLa cells because of the activity of IFN-α in inhibiting its proliferation (apoptosis and antiproliferative activity of interferons) [151]. Cells were seeded in 96-well plates (Corning Costar, Thermo Fisher Scientific, Waltham, MA, USA) at 10^4^ cells/mL in DMEM + 10% FBS + neomycin (100 IU/mL) and cultured at 37 °C with 5% CO_2_ for 24 h. Eight 1:5 serial dilutions of purified rhIFNα-2b, rhIFN-γ, and their respective standards were prepared in DMEM + SFB 10%, starting with a concentration of 200 ng/mL. Overall, 100 µL per well was added to the plates seeded with the cells. These were incubated at 37 °C and 5% CO_2_ for 24 h. Wells with untreated cells were used as a 100% cell viability control. After incubation, the plates were washed with phosphate-buffered saline (PBS), fixed with 100% methanol at 4 °C for 10 min, stained with a 0.5% crystal violet solution in 20% methanol for 15 min with gentle agitation, and the excess stain was removed by successive washes with water. The RFU was measured using the Synergy HTX Multimode Reader (Agilent Technologies Inc., Santa Clara, CA, USA). The plate was left to dry at room temperature and was finally scanned and read at 623 nm on the Odyssey CLx Imaging System (LI-COR Biosciences, Lincoln, NE, USA). The data were analyzed and graphed using GraphPad Prism v5.0 software (GraphPad Software, Boston, MA, USA) [88].

##### Evaluation of the Immunoregulatory Activity of Purified rhIFN-γ

The immunomodulatory activity of rhIFN-γ was analyzed in the COLO-320 cell line. COLO-320 cells in contact with rhIFN-γ express the MHC class II isotype DR antigen (HLA-DR II), which they do not structurally possess [89]. Cells were seeded in 6-well plates (Corning Costar, Thermo Fisher Scientific, Waltham, MA, USA) at 3 × 10^5^ cells/well in RPMI + 10% FBS and cultured at 37 °C with 5% CO_2_ for 24 h. Cells were treated with different concentrations of purified rhIFN-γ and its standard (0, 12.5, 25, 50, and 100 ng/mL) for. h at 37 °C with 5% CO_2_. The total protein lysates were recovered with radioimmunoprecipitation (RIPA) buffer (Pierce Biotechnology, Rockford, IL, USA) and a mixture of protease inhibitors (Halt^TM^ Protease Inhibitor Cocktail, EDTA-free, Pierce Biotechnology, Rockford, IL, USA). Protein quantification was performed using the bicinchoninic acid (BCA) method at 562 nm, as described in the Micro BCA protein assay kit (Thermo Fisher Scientific, Waltham, MA, USA) protocol, according to the manufacturer’s instructions. The absorbance was measured with the Synergy HTX Multimode Reader (Agilent Technologies Inc., Santa Clara, CA, USA). 10% SDS-PAGE separated protein samples under reducing conditions. The electrophoretic separation was performed in a Mini-PROTEAN Tetra System electrophoresis chamber (Bio-Rad Laboratories, Hercules, CA, USA) using Tris-Glycine buffer in both the cathodic and anodic compartments, at a constant current of 100 V for 1.5 h. Subsequently, the proteins were transferred to a nitrocellulose membrane by semi-dry electrotransfer using the Trans-Blot Turbo Transfer System (Bio-Rad Laboratories, Hercules, CA, USA) with transfer buffer at 250 mA for 30 min. The transfer efficiency was verified by staining with a 1% Ponceau solution for 5 min. The membrane was blocked with 5% (*w*/*v*) skim milk in Tris-buffered saline (TBS) for two hours at room temperature. Then, it was washed with TBS-T buffer (0.1% SDS, 0.1% Tween 20) and incubated with a monoclonal anti-HLA-DR antibody (Sigma-Aldrich, St. Louis, MO, USA) and a primary anti-GAPDH antibody (Thermo Fisher Scientific, Waltham, MA, USA) as a loading control. Detection was performed using a secondary anti-mouse IgG antibody conjugated with Alexa Fluor 680 (Jackson ImmunoResearch Inc., West Grove, PA, USA), which was incubated at room temperature with gentle agitation. The immunological signals were visualized using an infrared imaging system, the Odyssey CLx Imaging System (LI-COR Biosciences, Lincoln, NE, USA). The obtained data were processed and analyzed with the GraphPad Prism v5.0 software (GraphPad Software, Boston, MA, USA).

##### Stability Under Accelerated Conditions of Purified rhIFNα-2b and rhIFN-γ

The stability of rhIFNα-2b and rhIFN-γ in solution was evaluated under accelerated conditions by incubating the recombinant interferon samples at various temperatures: 4 °C, 16 °C, 25 °C, 30 °C, and 37 °C for 18 days using individual incubators (INDUCELL 55eco, BMT Medical Technology s.r.o., Brno, Czech Republic) for each thermal condition [192]. The antiviral activity of both proteins was evaluated following the protocol described in Section Antiviral Activity of Purified rhIFNα-2b and rhIFN-γ [188,189]. The data were fitted to a sigmoid curve, and the EC_50_ value was determined. Using IFNα-2b and IFN-γ standards as reference, interferon titer, specific activity, and concentration were determined for the purified proteins.

##### Statistical Analysis of In Vitro Studies

Statistical analysis was performed with SPSS v25.0, STATISTICA v6.0, and GraphPad Prism v9.0 software (GraphPad Software, Boston, MA, USA).


Antiproliferative activity of rhIFNα-2b and rhIFN-γ


The Shapiro–Wilk test was used to confirm that the normalized RFU data fit a normal distribution. An ANOVA test and a multiple comparisons test were used to verify which concentrations significantly affected cell viability.


Immunoregulatory Activity of rhIFN-γ


Data were adjusted to a normal distribution using the Shapiro–Wilk test. A two-way ANOVA test was performed to investigate the relationship between the different concentrations of the two proteins, and a pairwise comparison was conducted, taking into account the Bonferroni correction to control for Type I error. The statistical significance was set at α = 0.05.

#### 4.2.3. Endotoxin Analysis (Limulus Amebocyte Lysate Test)

Endotoxin levels in the purified interferon sample were determined using the Pierce^TM^ Chromogenic Endotoxin Quant Kit (Thermo Fisher Scientific, Waltham, MA, USA), following the manufacturer’s protocol. All materials in contact with the sample, including pipette tips, tubes, and microplates, were certified endotoxin-free. Samples and standards (0.01–1.0 EU/mL) were incubated with LAL reagent and a chromogenic substrate at 37 °C. The reaction stopped with 25% acetic acid, and absorbance was measured at 405 nm. Endotoxin concentrations were calculated by interpolation from a standard curve using linear. Results were reported in EU/mL.

#### 4.2.4. Animal Safety Studies with rhIFNα-2b and rhIFN-γ

##### LD_50_ with rhIFNα-2b and rhIFN-γ in Mice

The safety range is greater when it is further from the therapeutic dose and the minimum toxic dose, the interval at which the first signs of toxicity appear. The LD_50_ is an indicator of the acute toxicity of pharmaceutical products, corresponding to the dose that causes the death of 50% of the analyzed animals [193]. BALB/c mice of both sexes from the ISP, weighing between 24 and 28.5 g, were divided into four groups of 8 animals each, totaling 32 animals. The randomly distributed animals were maintained in controlled environmental conditions: a temperature of 22 ± 2 °C, a relative humidity of 55–65%, and a 12 h light:12 h dark lighting cycle. They were maintained on pelletized feed and water, “ad libitum”.

The experimental design consisted of administering a single dose of 50 µL via intraperitoneal injection to each group: Group I, the control with sterile physiological saline; Group II, with rhIFNα-2b at 1 × 10^7^ IU/kg; Group III, with rhIFNα-2b at 3 × 10^7^ IU/kg; and Group IV, with rhIFN-γ at 1 × 10^7^ IU/kg. Prior to the assay, the animals were housed in the acclimatization area for 7 days and individually identified by their exact dosage based on weight. The animals were observed for 14 days, with changes in behavior and possible signs of toxicity recorded every two days. At the end of the study, the animals were weighed and then sacrificed by inhalation of CO_2_ at a displacement rate of 30%, followed by cervical dislocation. The effects of the treatment on the parenchymal tissues were subsequently evaluated.

##### Cardiorespiratory Toxicity with rhIFNα-2b and rhIFN-γ in Rats

The evaluation of cardiorespiratory toxicity was performed in Sprague Dawley rats. The animals were divided into three groups of 5 animals each, for a total of 15 rats [167]. The experimental design consisted of administering a single dose of 1.8 × 10^7^ IU/kg of rhIFNα-2b or rhIFN-γ via intraperitoneal injection (150 µL) to each group: Group I, the control group with physiological saline; Group II, the rhIFNα-2b group; and Group III, the rhIFN-γ group. The baseline rhythm of each animal was recorded, including heart rate in beats per minute (bpm) and respiratory rate in breaths per minute (rpm), and these measurements were compared at 1, 5, 10, 20, and 30 min post-administration. After 72 h, signs of toxicity, including behavioral or neurological alterations, were evaluated; on the last day, the animals were weighed and sacrificed by inhalation of CO_2_ at a displacement rate of 30%, followed by cervical dislocation.

##### Subchronic Toxicity Studies with rhIFNα-2b in Rats

The purpose of these studies was to evaluate the toxic effects of repeated administration of a product at the selected therapeutic dose and two dose levels above it, for one month, in Sprague Dawley rats with body weights between 300 and 400 g, as provided by the ISP [9]. The rats were randomly divided into four groups of six animals, totaling 24 animals. The animals were maintained under good laboratory practices and underwent periodic clinical supervision and body weight measurements every week during the experiment.

The experimental design consisted of intraperitoneal administration twice (150 µL) per week for 28 days to each group: Group I, the control with saline solution; Group II, with rhIFNα-2b at 1 × 10^5^ IU/kg, Group III, with rhIFNα-2b at 1 × 10^6^ IU/kg; and Group IV, with rhIFNα-2b at 3 × 10^6^ IU/kg. Half of the animals in each group were sacrificed on day 28, and the rest of the animals 28 days after the drug was suspended. The criterion for the recovery of possible product actions was evaluated after one month of exposure. Additionally, blood chemistry was performed on all animals at the end of treatment and 28 days post-treatment. The animals were sacrificed by inhalation of CO_2_ at a displacement rate of 30%, followed by cervical dislocation. The kidneys, liver, spleen, thymus, heart, and lungs were weighed to evaluate the effects of the treatment on the parenchymal tissues. The blood sample was obtained by puncturing the abdominal aorta for blood chemistry determinations, and a macro- and micro-histological study was performed on the target organs and parenchymal tissues.

##### Pyrogen Study with rhIFNα-2b and rhIFN-γ in Rabbits

The study was conducted in accordance with the recommendations of ISO 10993-1 [181] and USP 42-NF 37 standards [182], which involve measuring the increase in body temperature in rabbits after sample administration [194]. Clinically healthy adult rabbits were randomly divided into five groups of two animals, totaling 10 animals, with a body mass ranging from 3.8 to 4.4 kg, obtained from the Faculty of Veterinary Medicine at the University of Concepción, Chillán campus. Rabbits that lost more than 5% of their body mass during the acclimatization period (under the same conditions as the assay) were discarded. Weight measurements were recorded at the beginning, 7 days after the start of the assay, and at the end of the treatment. The animals’ body temperature was also recorded one week before the treatment, a variable that oscillated within the normal range (36–39 °C). There was no need to withdraw animals from the study with temperatures below 37.8 °C and/or above 39.8 °C.

The experimental design consisted of a single dose (500 µL) to each group: Group I, the control (untreated); Group II, placebo with saline solution; Group III, with rhIFNα-2b at 1.5 × 10^5^ IU/kg; Group IV, with rhIFNα-2b at 4 × 10^5^ IU/kg; and Group V, with rhIFN-γ at 1.5 × 10^5^ IU/kg. The rabbits were fed until 12 h before the start of the assay and were weighed before being placed in handling boxes. Rectal temperatures were initially recorded and then measured every hour for the first 7 h, continuing daily until the end of the assay. The inoculation was performed in the marginal ear vein of each rabbit, with feeding resumed 30 min after inoculation. The animals were evaluated daily under baseline conditions, including feeding, mobility, presence of stigmata, and signs of decay. No animal sacrifice was required at the end of the experiment, as the study of organs and parenchyma was not part of the design.

##### Mucosal Toxicity Study of Interferons in Sheep

The study was designed to determine the safety of the active ingredients of two interferons in a higher-organism animal model, specifically by evaluating mucosal irritability in sheep [183]. Healthy adult sheep, all female, with a body mass between 37 and 64 kg, were selected. The sheep were randomly divided into four experimental groups of four animals each, totaling 16 animals. Each group received a single dose applied to the nostrils.

The animals were housed in the acclimatization area for 7 days prior to the assay and were individually identified according to their treatment group. They were housed in stalls with water “ad libitum” during the assay and were kept in an isolated room with a stainless-steel table. During the assay, light/dark cycles, humidity, and ambient temperature were maintained. The sheep were housed in stalls that were 2.4 m high, 2.5 m wide, and 2.5 m long, with a surface area of 1.5 to 2 m^2^ per animal. Cleaning was performed daily. The diet administered was 3% of live weight, with a 19% protein concentrate and clover hay. The experimental procedures with the animals were adequate to prevent suffering or pain, with daily observations and measurements to detect any alterations, stress, or signs of animal suffering that would have led to stopping the assay.


Experimental design:


Group I: Control. 4 untreated sheep received sterile saline solution (0.9% NaCl), 2 mL per nostril.

Group II: 4 sheep treated with the active ingredient rhIFNα-2b at a dose of 1 × 10^7^ IU/kg, 2 mL per nostril.

Group III: 4 sheep treated with the active ingredient rhIFN-γ at a dose of 1 × 10^6^ IU, 2 mL per nostril.

Group IV: 4 sheep treated with the active ingredients rhIFNα-2b (1 × 10^7^ IU/kg) and rhIFN-γ (1 × 10^6^ IU/kg), 2 mL of the solution per nostril.

This assay lasted one month to evaluate the safety of the formulation, supplemented with a histopathological study of the nasal mucosa. Samples of the nasal vestibule were taken by a 4 mm punch biopsy on days 4, 8, 12, and 16 from each sheep. The samples were obtained from one nostril, leaving the other nostril for subsequent evaluation, allowing for tissue recovery between sample collections. Before the biopsy, the animals were locally anesthetized with an infiltrative facial nerve block using 0.5 mL of Mepivacaine as an analgesic and 0.1 mg/kg of Xylazine 2% as a tranquilizer. The sample was stored in Eppendorf tubes containing a 10% buffered formaldehyde solution for histopathological study.

For the histopathological study, each sample was entirely placed in an inclusion cassette, and a dehydration process was performed using a series of ascending grades of alcohol (ethanol) (70, 80, 95, and 100%). The samples were cleared with xylene and embedded in paraffin using a Citadell 1000 tissue processor (Thermo Fisher Scientific, Waltham, MA, USA). Paraffin blocks were prepared using a Microm AP280-1 embedder (Wazobia Enterprise, Houston, TX, USA). The blocks were cut to a thickness of 4 μm using an RM2045 rotation microtome (Leica Biosystems, Nussloch, Germany) and stained with hematoxylin-eosin (Merck KGaA, Darmstadt, Germany), according to the standardized protocol of the Histopathology Laboratory of the Department of Pathology and Preventive Medicine at the Faculty of Veterinary Sciences, Universidad de Concepción. At the end of the study, the animals were not sacrificed, as this was deemed unnecessary due to the design of the sample collection.

##### Statistical Analysis of Animal Studies

For the statistical analysis of the LD_50_ study, cardiorespiratory toxicity, and pyrogen study, the variables were summarized by the mean and standard deviation. The normality of the variables was determined using the Shapiro–Wilk test. The treatment groups were compared for the variables of body weight, organ weight, heart and respiratory rate, and temperature with a one-way ANOVA test. The paired analysis for the comparison between times and treatment groups was performed using Student’s *t*-test for dependent samples.

In the case of the subchronic toxicity assay, the variables were summarized by the mean and standard deviation. The normality of the variables was determined using the Shapiro–Wilk test. To compare the body weight and organ weight variables between the Group 1 month of treatment and the 28-day recovery group, an independent samples *t*-test was used. For the blood chemistry variable, an independent samples Student’s *t*-test was used for the variables glutamic-pyruvic transaminase (GPT), proteins, hematocrit, and cholesterol. However, for the variables glutamic-oxaloacetic transaminase (GOT), hemoglobin, and urea, the non-parametric Mann–Whitney test was used as the Shapiro–Wilk test did not yield a normal distribution for these variables. The statistical significance was set at α = 0.05.

For the mucosal toxicity study in sheep, the mean and standard deviation variables were summarized in the statistical analysis. The normal distribution was determined for weight, temperature, and nasal mucosa status using the Shapiro–Wilk test. Treatment groups were compared using a one-way ANOVA for the variables body weight and temperature at baseline and at the end of the study, between the different treatment groups. We performed a correlation analysis using paired analysis between times for the variables body weight and temperature, employing repeated measures ANOVA. For the two-by-two comparison within each treatment group, we used a paired *t*-test. A Bonferroni error correction was considered, as there were more than two evaluations. The statistical significance level was set at α = 0.05.

All variables were analyzed using the statistical programs SPSS 25.0, STATISTICA 6.0, and GraphPad Prism v9.0.

## 5. Conclusions

The absence of systematic studies integrating the structural stability and intrinsic toxicity of rhIFNα-2b and rhIFN-γ represents a critical barrier to their preclinical characterization. Available models have focused on modified or combined variants, making it challenging to predict the effects attributable to the native molecules accurately. The findings presented in this work not only bridge this gap but also confirm the functionality and stability of interferons under controlled conditions, establishing a reproducible experimental model for future preclinical evaluations and serving as a basis for the clinical development of recombinant formulations, including potential alternative administration routes, such as intranasal.

This study’s approach highlights the importance of comprehensive preclinical research from the earliest stages. It was demonstrated that the molecules maintain their bioactivity (antiviral, antiproliferative, and immunomodulatory) and exhibit no signs of acute or subchronic toxicity in the evaluated models. This finding, along with the absence of pyrogens, lays the foundation for exploring advanced formulations and alternative administration routes, such as the nasal route, which, despite its potential, has been historically underestimated.

The success of this work serves as a guide for future biopharmaceutical developments, as it emphasizes the need to characterize the safety profile of the active ingredient before advancing with specific formulations, which will ensure a safe and well-founded transition to human clinical trials.

## Figures and Tables

**Figure 1 ijms-26-11982-f001:**
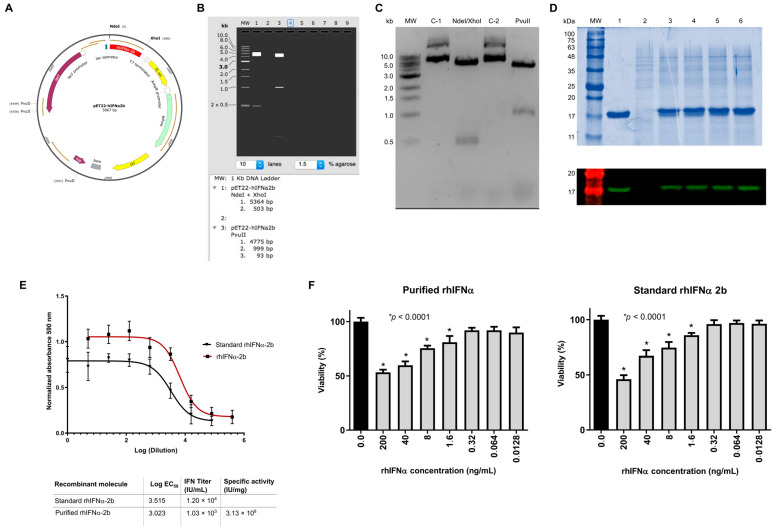
Characterization of the rhIFNα-2b protein. (**A**): Map of the pET22b-rhIFNα-2b vector, showing the location of the NdeI and XhoI, and PvuII restriction endonuclease cleavage sites. (**B**): In silico digestion of the pET22b-rhIFNα-2b vector. MW: 1 kb DNA molecular weight marker (New England BioLabs) 1: vector digested with NdeI, XhoI. 3: vector digested with PvuII. (**C**): 1% agarose gel electrophoresis of the pET22b-rhINFa-2b vector digested with the NdeI and XhoI, and PvuII restriction endonucleases. MW: 1 kb DNA molecular weight marker (New England BioLabs) NdeI/XhoI: vector digested with NdeI, XhoI. PvuII: vector digested with PvuII. C-1 and C-2: undigested vector. (**D**): 15% SDS-PAGE and Western blot immunodetection of *E. coli* SHuffle^®^ T7 Express clones expressing rhIFNα-2b. MW: Molecular weight pattern (AccuRuler RGB Maestrogen, Hsinchu, Taiwan). 1: Positive control of standard rhIFNα-2b (Laboratorios Bagó). 2: Negative control of *E. coli* SHuffle^®^ T7 Express. 3–6: *E. coli* SHuffle^®^ T7 Express/pET22b-rhIFNα-2b clone. (**E**): Antiviral activity: titer and specific activity of purified rhIFNα-2b and standard rhIFNα-2b. (**F**): Antiproliferative activity of purified and standard rhIFNα-2b at different dilutions in HeLa cells. ANOVA and multiple comparisons tests were used to evaluate cell viability. *: *p* < 0.0001.

**Figure 2 ijms-26-11982-f002:**
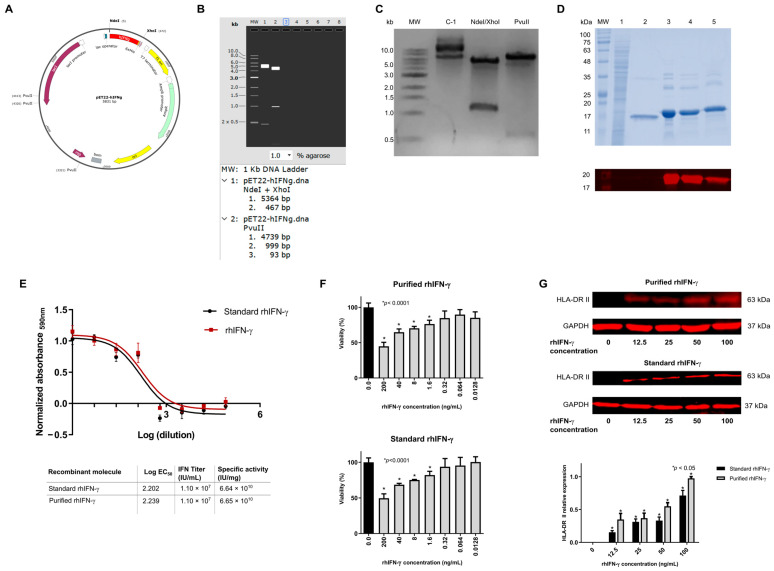
Characterization of the rhIFN-γ protein. (**A**): Map of the pET22b-rhIFN-γ vector, showing the location of the NdeI and XhoI, and PvuII restriction endonuclease cleavage sites. (**B**): In silico digestion of the pET22b-rhIFN-γ vector. MW: 1 kb DNA molecular weight marker (New England BioLabs) 1: vector digested with NdeI, XhoI. 3: vector digested with PvuII. (**C**): 1% agarose gel electrophoresis of the pET22b-rhINFa-2b vector digested with the NdeI and XhoI, and PvuII restriction endonucleases. MW: 1 kb DNA molecular weight marker (New England BioLabs) NdeI/XhoI: vector digested with NdeI, XhoI. PvuII: vector digested with PvuII. C-1: undigested vector. (**D**): 15% SDS-PAGE and Western blot immunodetection of *E. coli* SHuffle^®^ T7 Express clones expressing rhIFN-γ. MW: Molecular weight pattern (AccuRuler RGB Maestrogen, Hsinchu, Taiwan). 1: Positive control of standard rhIFN-γ (Laboratorios Bagó). 2: Negative control of *E. coli* Shuffle^®^ T7 Express. 3–5: *E. coli* SHuffle^®^ T7 Express/pET22b- rhIFN-γ clone. (**E**): Antiviral activity: titer and specific activity of purified rhIFN-γ and standard rhIFN-γ. (**F**): Antiproliferative activity of purified and standard rhIFN-γ at different dilutions in HeLa cells. ANOVA and multiple comparisons tests were used to evaluate cell viability. *: *p* < 0.0001. (**G**): Analysis of the expression of the HLA-DR II antigen in COLO-320 cells treated with rhIFN-γ by Western blot. A primary anti-HLA-DR monoclonal antibody and a secondary anti-mouse IgG antibody conjugated to Alexa Fluor 680 were used. Cells were treated at different rhIFN-γ concentrations: 0, 12.5, 25, 50, and 100 ng/mL.

**Figure 3 ijms-26-11982-f003:**
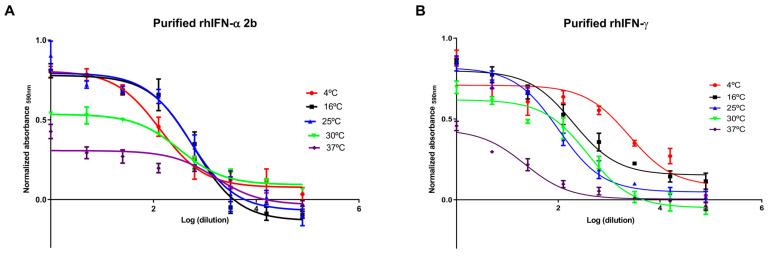
Accelerated stability at various temperatures of purified rhIFN-α-2 b and rhIFN-γ. Evaluation of viral activity by the cytopathogenic effect produced by the Mengo virus in Hep-2 cells. Log (dilution) graph of recombinant rhIFN-γ, to which Hep-2 cells were exposed to counteract the cytotoxic effect of the Mengo virus as a function of absorbance at 590 nm, normalized with respect to the cell control (100% viability) and virus control (maximum mortality). (**A**): Log (dilution) graph of recombinant rhIFNα-2b. (**B**): Log (dilution) graph of recombinant rhIFNα-2b.

**Figure 4 ijms-26-11982-f004:**
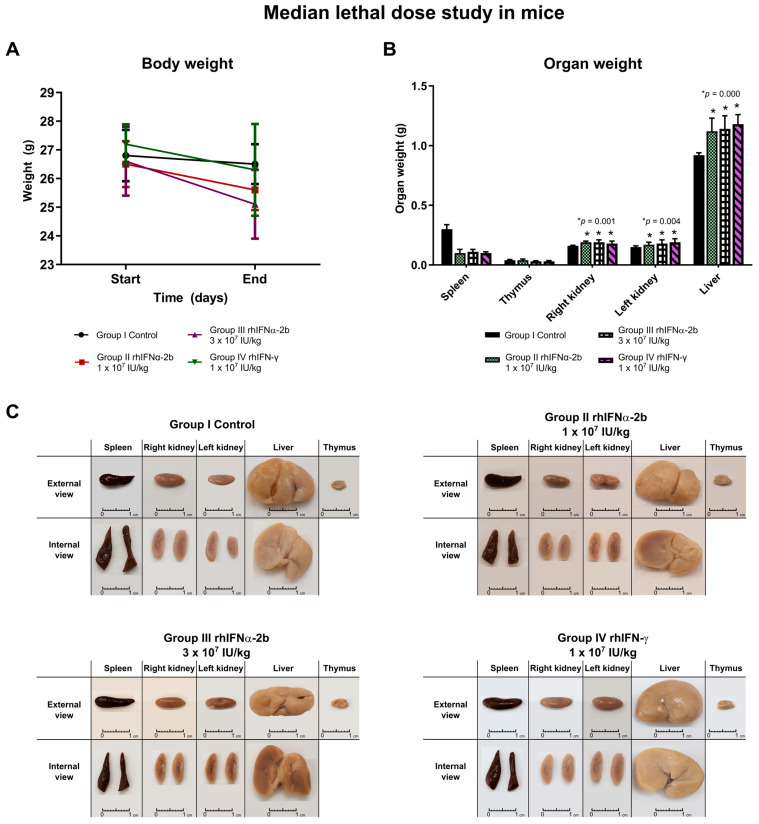
Median lethal dose study in mice. (**A**): Weight of organs (g) (*n* = 8) according to applied treatments. (**B**): Weight of animals at the beginning and end of the assay. (**C**): Photographs of the vital organs of the animals according to applied treatments. The mice (*n* = 8) assigned to the different treatment groups received various doses of rhIFN-α-2 b. Fourteen days post-treatment, the animals were sacrificed, and the weights of the other organs were evaluated. The average organ weight is graphed. The treatment groups were compared through a one-way ANOVA and a Student’s *t*-test for dependent samples. The statistical significance was set at α = 0.05.

**Figure 5 ijms-26-11982-f005:**
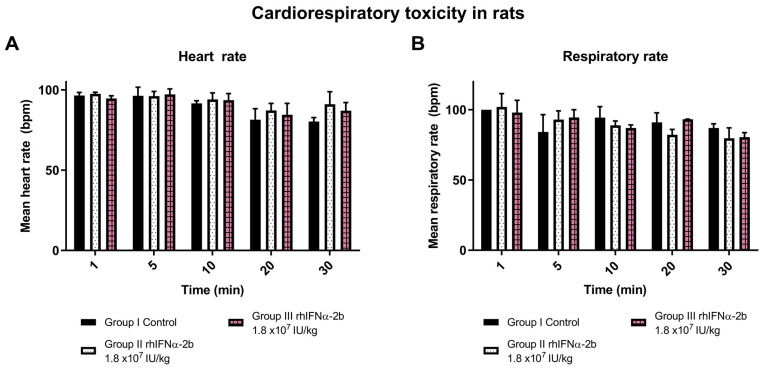
Evaluation of cardiorespiratory toxicity in rats. The animals (n = 5) assigned to the different treatment groups received a single dose of rhIFNα-2b and rhIFN-γ via the intraperitoneal route, according to their assigned group. Basal rhythm and respiratory rate were measured. (**A**): Mean heart rate for each treatment. (**B**): Mean respiratory rate for each treatment. The treatment groups were compared through a one-way ANOVA and a Student’s *t*-test for dependent samples. The statistical significance was set at α = 0.05.

**Figure 6 ijms-26-11982-f006:**
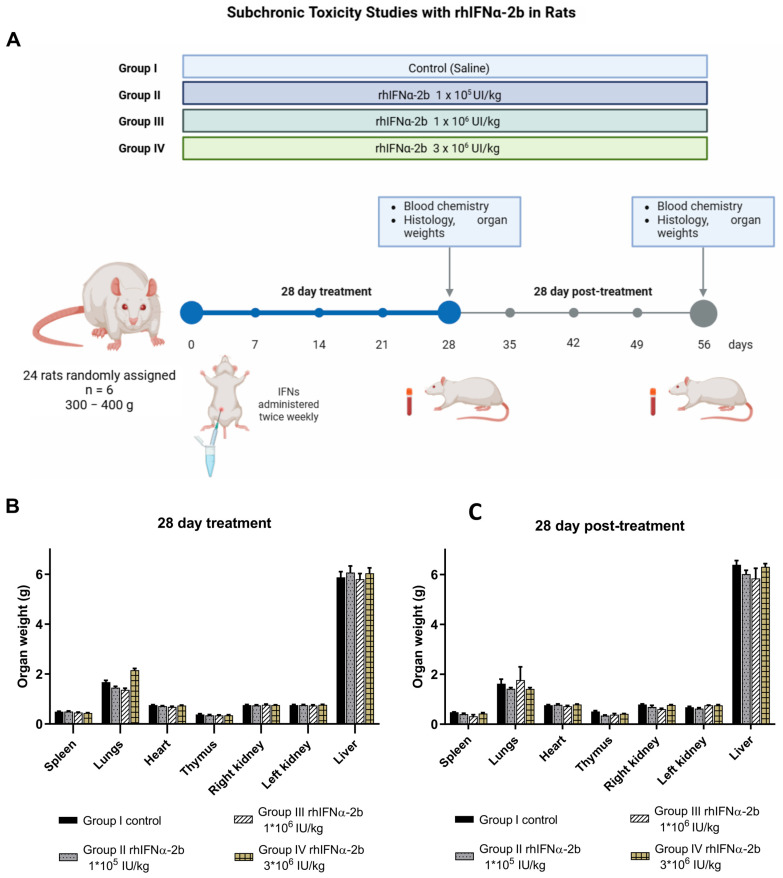
Evaluation of subchronic toxicity in rats. The animals (n = 6) assigned to the different treatment groups received various doses of rhIFNα-2b, administered intraperitoneally twice a week for 28 days, followed by a recovery period of one month. Blood chemistry and organ weight were evaluated. The relationship between organ weight and organ type was evaluated. (**A**): Experimental design of subchronic toxicity. Created with Biorender.com. (**B**): Organ weights of the animals after 28 days of treatment. (**C**): Organ weights of the animals after 28 days post-treatment. The treatment groups were compared through a Student’s *t*-test. The statistical significance was set at α = 0.05.

**Figure 7 ijms-26-11982-f007:**
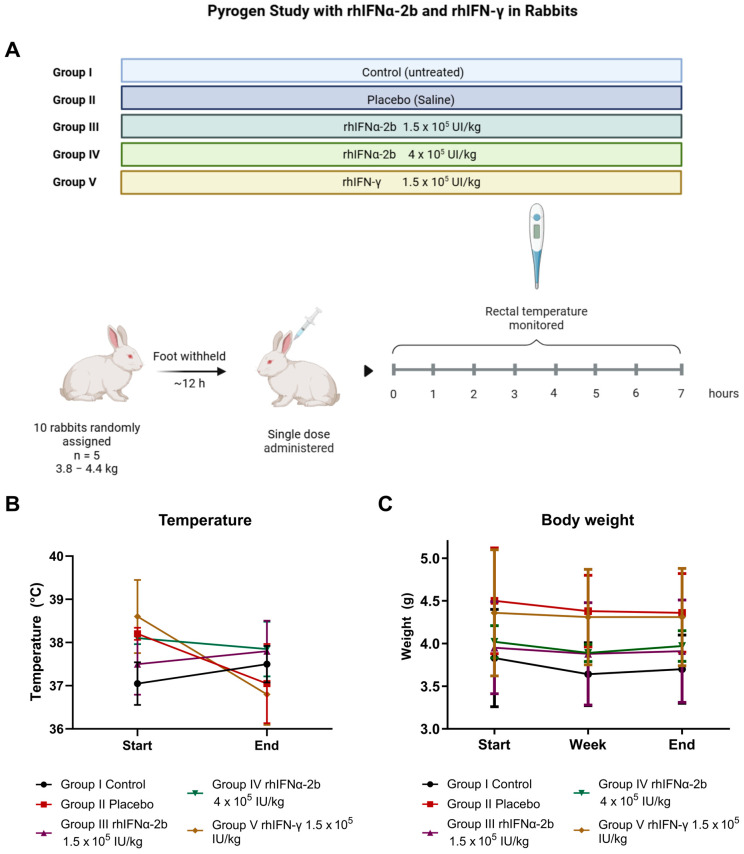
Pyrogen Study in Rabbits. (**A**): Experimental design of the pyrogen study. Created with Biorender.com. (**B**): Temperature fluctuations (°C) from start to end of the experiment according to treatment group. (**C**): Relationship between body weight (g) and time for each treatment group. The rabbits (n = 2) assigned to the different treatment groups received various doses of rhIFN-α-2b and rhIFN-γ, and their body weights were recorded. The average change in body weight at multiple times is graphed. Data are expressed as mean ± SD (three replicates). The treatment groups were compared through a one-way ANOVA and a Student’s *t*-test for dependent samples. The statistical significance was set at α = 0.05. There were no significant differences between the groups.

**Figure 8 ijms-26-11982-f008:**
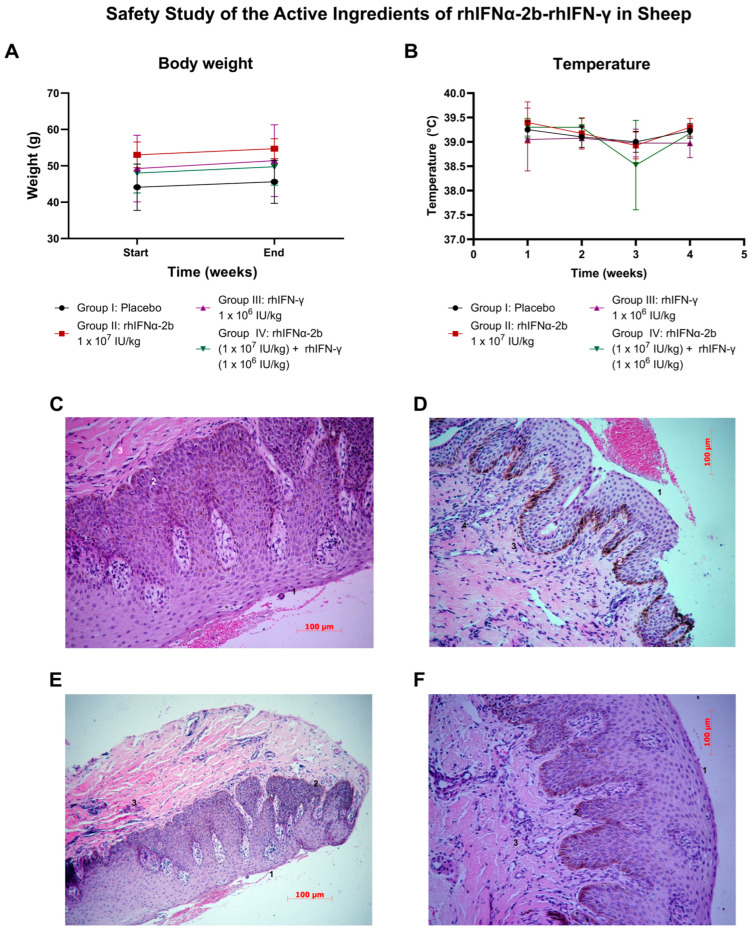
Safety Study of the Active Ingredients of rhIFNα-2b-rhIFN-γ in Sheep. (**A**): Body weight fluctuations (°C) from start to end of the experiment according to treatment group. Data are expressed as mean ± SD (three replicates). Treatment groups were compared through a one-way ANOVA. The statistical significance level was set at α = 0.05. (**B**): Relationship between body weight (g) and time for each treatment group. Data are expressed as mean ± SD (three replicates). Treatment groups were compared through a one-way ANOVA. The statistical significance level was set at α = 0.05. (**C**): Histopathological study, Group I: Placebo. Nasal vestibule sample, showing slightly keratinized stratified epithelium (1) with abundant melanocytes (2). Lamina propria and connective tissue preserved in appearance (3). (**D**): Histopathological study, Group II: rhIFNα-2b 1 × 107 IU/kg. Nasal vestibule sample, stratified epithelium with sparse keratinization (1) and abundant melanocytes (2). Lamina propria of normal appearance (3). Mild leukocyte infiltrate (4). (**E**): Histopathological study, Group III: rhIFN-γ- γ 1 × 10^6^ IU/kg. Nasal vestibule sample, stratified epithelium with sparse keratinization (1) and abundant melanocytes (2). Mild leukocyte infiltrate (3). (**F**): Histopathological study, Group IV: rhIFNα-2b (1 × 10^7^ IU/kg) + rhIFN-γ- γ (1 × 10^6^ IU/kg). Nasal vestibule sample, stratified epithelium with sparse keratinization (1) and abundant melanocytes (2). Mild leukocyte infiltrate (3).

**Table 1 ijms-26-11982-t001:** LD_50_ study. Changes in body weight (g) of the animals according to the dose used per group.

Body Weight/Treatment		Group I Control	Group II rhIFNα-2b 10,000,000 IU/kg	Group III rhIFNα-2b 30,000,000 IU/kg	Group IV rhIFN-γ 10,000,000 IU/kg	ANOVA F (*p*)
Total		8	8	8	8	
Start/Mean ± SD		26.8 ± 0.9	26.5 ± 0.8 *	26.6 ± 1.2 *	27.2 ± 0.7	0.952 (0.429)
End/Mean ± SD		26.5 ± 0.7	25.6 ± 0.7 *	25.1 ± 1.2 *	26.3 ± 1.6	2.526 (0.078)
*p* (Student’s t)	Start vs. End	0.108	0.006	0.004	0.059	

* The table represents the mean ± standard deviation (SD). The treatment groups were compared through a one-way ANOVA and a Student’s *t*-test for dependent samples. The statistical significance was set at α = 0.05.

## Data Availability

The original contributions presented in the study are included in the article/Appendix A, further inquiries can be directed to the corresponding author.

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
