# Peer review of "From Genetic Engineering to Preclinical Safety: A Study on Recombinant Human Interferons"

_ijms, 2025, doi:10.3390/ijms262411982_

Round 1
Reviewer 1 Report
Comments and Suggestions for Authors
This paper is reporting relatively sound work to produce recombinant interferons for human use, but it misrepresenting the field and the relevance of their own impact. It is not fair to claim that previous work has focused on modifications or interferons, such as PEGylation, since many decades of work since the discovery of interferons have thoroughly documented the biological functions and properties of interferons. Instead, work to utilize interferons for human immune effects have concluded that natural forms of interferons have a much too short half-life to be practical. Therefore, various half-life prolonging strategies have been explored.
The authors will need to refocus on this reality rather than disparaging such efforts to make pharmaceutically useable interferon therapeutics.
Comments on the Quality of English LanguageThe language is rather complex and convoluted and could be much improved for clarity and brevity.
Reviewer 2 Report
Comments and Suggestions for Authors
The paper by Ramos et al. thoroughly assesses the structural stability and inherent toxicity of purified IFNs. This study may serve as a valuable supplement to similar published research. I have a few comments and questions listed below:
- Would it be possible to include a summary table highlighting the key findings across measured parameters for both IFNs, particularly in terms of potency, cellular and functional effects, and toxicity?
- rhIFNα‐2b is more potent as an antiviral but tends to be more immunogenic systemically, whereas rhIFN‐γ is less broadly active yet plays a critical role in immune activation and antifibrotic regulation. Are there any species-specific receptor interactions that significantly influence the relative efficacy or toxicity of these IFNs in different contexts?
- Would it be feasible to incorporate transcriptomic analyses of immune cells following treatment with either rhIFNα‐2b or rhIFN‐γ, including considerations of species differences?
Round 2
Reviewer 1 Report
Comments and Suggestions for Authors
I am satisfied with the author's response and amendment of the paper.